EMBO
Molecular Medicine

# Prevention of excitotoxicity-induced processing of BDNF receptor TrkB-FL leads to stroke neuroprotection

Gonzalo S Tejeda[†,‡] ⓘ, Gema M Esteban-Ortega[‡], Esther San Antonio, Óscar G Vidaurre & Margarita Díaz-Guerra[*] ⓘ

## Abstract

Neuroprotective strategies aimed to pharmacologically treat stroke, a prominent cause of death, disability, and dementia, have remained elusive. A promising approach is restriction of excitotoxic neuronal death in the infarct penumbra through enhancement of survival pathways initiated by brain-derived neurotrophic factor (BDNF). However, boosting of neurotrophic signaling after ischemia is challenged by downregulation of BDNF high-affinity receptor, full-length tropomyosin-related kinase B (TrkB-FL), due to calpain-degradation, and, secondarily, regulated intramembrane proteolysis. Here, we have designed a blood–brain barrier (BBB) permeable peptide containing TrkB-FL sequences (TFL$_{457}$) which prevents receptor disappearance from the neuronal surface, early induced after excitotoxicity. In this way, TFL$_{457}$ interferes TrkB-FL cleavage by both proteolytic systems and increases neuronal viability via a PLCγ-dependent mechanism. By preserving downstream CREB and MEF2 promoter activities, TFL$_{457}$ initiates a feedback mechanism favoring increased levels in excitotoxic neurons of critical prosurvival mRNAs and proteins. This neuroprotective peptide could be highly relevant for stroke therapy since, in a mouse ischemia model, it counteracts TrkB-FL downregulation in the infarcted brain, efficiently decreases infarct size, and improves neurological outcome.

**Keywords** cell-penetrating peptides; excitotoxicity; neuroprotection; stroke; TrkB

**Subject Categories** Neuroscience; Pharmacology & Drug Discovery; Vascular Biology & Angiogenesis

## Introduction

Stroke is a leading cause of death, disability, and dementia. In the ischemic type, treatments are largely limited to mechanical strategies or pharmacological therapy with thrombolytic drugs. The decrease in brain perfusion due to vessel occlusion results in formation of the infarct core, irreversibly damaged tissue surrounded by an area of penumbra that is functionally impaired but metabolically active. However, infarct frequently expands to this area due to secondary neuronal death caused by overstimulation of the *N*-methyl-ᴅ-aspartate type of glutamate receptors (NMDARs) and consequent excitotoxicity. This damage is also associated with other acute and chronic CNS disorders (Choi, 1988), adding relevance to the development of excitotoxicity therapies. A promising approach is enhancement of survival pathways such as those regulated by brain-derived neurotrophic factor (BDNF) through binding to tropomyosin-related kinase B (TrkB) receptors. Remarkably, BDNF/TrkB potentiation would be protective in acute stroke and stimulate neurorepair at later stages (Berretta *et al*, 2014).

Promotion of neuronal survival by BDNF is mediated by dimerization, increased tyrosine kinase (TK) activity, and transphosphorylation of its high-affinity receptor full-length TrkB (TrkB-FL). Activation triggers three interconnected cascades: MAPK/ERK (mitogen-activated protein kinase/extracellular signal-regulated kinase), PI3K/Akt (phosphatidylinositol 3-kinase/v-Akt murine thymoma viral oncogene homolog), and PLCγ (phospholipase Cγ; Huang & Reichardt, 2003; Reichardt, 2006). Among other functions, these cascades activate prosurvival transcription factors (TFs) CREB (cAMP response-element binding protein; Bonni *et al*, 1999) and MEF2 (myocyte enhancer factor 2; Liu *et al*, 2003) that regulate the expression of target genes, including those coding BDNF (Tao *et al*, 1998; Lyons *et al*, 2012), TrkB (Deogracias *et al*, 2004) or NMDAR subunits (Krainc *et al*, 1998; Desai *et al*, 2002; Lau *et al*, 2004). Neurons also express TrkB-T1, a truncated isoform lacking the TK and having a short cytoplasmic domain

Instituto de Investigaciones Biomédicas "Alberto Sols", Consejo Superior de Investigaciones Científicas-Universidad Autónoma de Madrid (CSIC-UAM), Madrid, Spain
*Corresponding author. Tel: +34 91 5854443; E-mail: mdiazguerra@iib.uam.es
‡These authors contributed equally to this work
†Present address: Gardiner Laboratory, Institute of Cardiovascular and Medical Sciences, College of Medical, Veterinary and Life Sciences, University of Glasgow, Glasgow, UK

with an isoform-specific sequence. It is considered a dominant-negative receptor that inhibits BDNF signaling by forming hetero-dimers with TrkB-FL, and sequestering and translocating BDNF (Haapasalo *et al*, 2001). Upon binding, both isoforms are rapidly and efficiently internalized in a clathrin-dependent way (Zheng *et al*, 2008; Huang *et al*, 2009) and form signaling endosomes (Huang *et al*, 2009). While TrkB-T1 predominantly recycles back to cell surface by a default mechanism, TrkB-FL recycling is less efficient, relies on TK activity, and is regulated by binding of Hrs (hepatocyte growth factor-regulated tyrosine kinase substrate) to a juxtamembrane region between the transmembrane (TM) and TK domains (Huang *et al*, 2009).

BDNF/TrkB signaling becomes profoundly aberrant in stroke (Berretta *et al*, 2014; Tejeda & Diaz-Guerra, 2017). A permanent BDNF decrease is found in the infarct core, while a long-lasting upregulation, proposed to be a neuroprotective mechanism, takes place in the penumbra (Ferrer *et al*, 2001; Madinier *et al*, 2013). However, other studies rejected BDNF involvement in post-stroke recovery (Hirata *et al*, 2011), a likely explanation being pathological downregulation of BDNF receptors. Three independent mechanisms induced by excitotoxicity act on TrkB: (i) An inversion of mRNA ratios disfavors TrkB-FL expression over TrkB-T1 (Gomes *et al*, 2012; Vidaurre *et al*, 2012); (ii) TrkB-FL cleavage by $Ca^{2+}$-dependent calpain generates a truncated receptor similar to TrkB-T1 (Gomes *et al*, 2012; Vidaurre *et al*, 2012); and (iii) regulated intramembrane proteolysis (RIP) of both isoforms by metalloproteinase/γ-secretase action sheds identical ectodomains acting as BDNF scavengers (Tejeda *et al*, 2016). While RIP highly contributes to TrkB-T1 downregulation in ischemia, it is only a secondary mechanism for TrkB-FL, mainly processed by calpain. The importance of calpain for TrkB-FL dysregulation in excitotoxicity-associated disorders other than stroke has been confirmed in epilepsy (Danelon *et al*, 2016) and Alzheimer disease (AD; Jeronimo-Santos *et al*, 2015). Calpain activation (Adamec *et al*, 2002) and TrkB-FL loss (Allen *et al*, 1999; Ferrer *et al*, 1999) had been observed before in the brain of AD patients, while a TrkB-FL calpain processing site was mapped by Edman sequencing in neurons treated with Aβ (Jeronimo-Santos *et al*, 2015). Altogether, the mechanisms described impair BDNF signaling and point to TrkB as a therapeutic target for neuroprotection in disorders associated with excitotoxicity.

To preserve BDNF-regulated survival pathways, we have designed and characterized cell-penetrating peptides (CPPs) based on the BDNF receptor. They contain a Tat sequence, which allows attached cargoes to cross the blood–brain barrier (BBB) and plasma membrane (Regberg *et al*, 2013), and different TrkB-FL sequences that we hypothesized might control receptor stability and function in excitotoxicity. One peptide (TFL$_{457}$) proved to interfere TrkB-FL calpain processing and RIP in rat primary neurons subjected to *in vitro* excitotoxicity. The primary mechanism of TFL$_{457}$ action is to maintain TrkB-FL in the cell surface, apart from the proteolytic machinery activated in excitotoxicity. The preserved receptor triggers a feedback survival mechanism mediated by PLCγ, and CREB and MEF2 promoter activities. Importantly, this neuroprotective peptide also counteracts TrkB-FL downregulation in mouse ischemia, where it efficiently decreases infarct size and improves neurological outcome, unraveling a highly relevant strategy for stroke therapy.

# Results

## TrkB-FL juxtamembrane region is a rational target for the design of CPPs interfering receptor downregulation

To approach the design of CPPs able to preserve TrkB-FL levels in excitotoxicity and ischemia, we focused in the receptor inter-domain sequence located between the TM and proximal TK domains (aa 453–536) for several reasons. Firstly, a calpain cleavage site had been mapped between residues N520/S521 in this region in an AD model (Fig 1A, red arrow; Jeronimo-Santos *et al*, 2015). Calpains often proteolyze a limited number of specific sites inside substrates inter-domains and, thus, additional processing sites in TrkB-FL could not be excluded. Secondly, sequences in the juxtamembrane region have been shown to be important for the regulation of different aspects of TrkB-FL location and function via protein interaction (Huang *et al*, 2009, 2011; Guo *et al*, 2017; Zamani *et al*, 2018). Lastly, we found an intrinsically disordered region (IDR) inner to this TrkB-FL juxtamembrane sequence (Fig EV1; Kozlowski & Bujnicki, 2012). IDRs are considered central units of protein function and regulation due to their ability to establish multiple interactions (Dunker *et al*, 2008) and their potential to act as weak signals for degradation (Tompa *et al*, 2008). Interestingly, *in silico* analysis of this TrkB-FL region using several predictive algorithms for calpain processing [CAMPDB (DuVerle *et al*, 2011), GPS-Calpain Cleavage Detector (Liu *et al*, 2011) and SitePrediction (Verspurten *et al*, 2009)] showed that predictions made by at least two algorithms mostly clustered in two areas (Fig 1A, black arrows) inside the described IDR (Fig EV1). Three of them were found nearby the TM in residues shared by all isoforms (light blue) and of potential functional importance according to a heatmap representation of all possible mutations (Hecht *et al*, 2015). We also found a second cluster of calpain sites in this same IDR and two more predictions in the proximal TK region.

Next, we selected four sequences potentially important for TrkB-FL regulation (Fig 1A, black rectangles) and generated CPPs containing them fused to a HIV-1 Tat basic domain, which confers membrane permeability and the capability of crossing the BBB to attached cargoes (Regberg *et al*, 2013; Fig 1B). These CPPs contained, respectively, TrkB-FL aa 457–471 (TFL$_{457}$), 482–495 (TFL$_{482}$), 518–531 (TFL$_{518}$), and 541–555 (TFL$_{541}$). As a negative control, we designed a similar Tat peptide with unrelated sequences corresponding to c-Myc (Fig 1B, TMyc). We verified TMyc capability to cross the membrane by incubating primary cortical cultures with biotin-labeled TMyc (Bio-TMyc; Fig 1C, panel b). Compared to untreated cultures (panel a), Bio-TMyc was detected in the cell body and neurites of the majority of neurons present in the culture, identified by NeuN (panel b, arrowheads). Quantitation of peptide entry showed that $83 \pm 4\%$ ($n = 5$) of neurons had internalized it (Fig EV2). TMyc suitability as a control peptide was also tested in neuronal viability assays. Treatment with NMDAR co-agonists, NMDA (100 μM) and glycine (10 μM; herein denoted NMDA), dramatically decreased neuronal viability which was not statistically modified by peptide preincubation, both in basal or excitotoxic conditions (Fig 1D). This result shows that Tat-based CPPs have no generic effects on neuronal viability and can be used to test potential neuroprotective sequences.

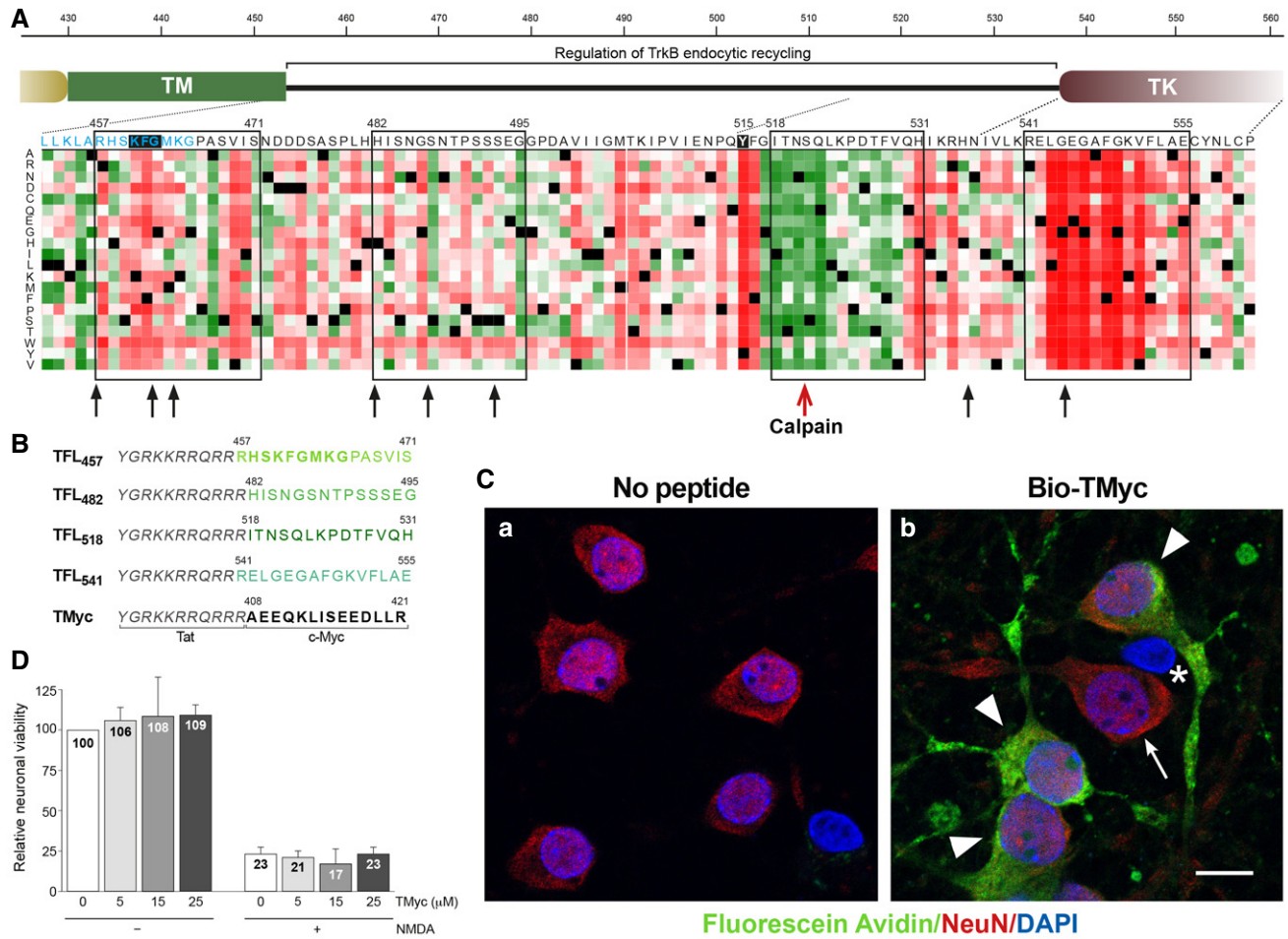

**Figure 1. TrkB-FL juxtamembrane region is a rational target for design of CPPs interfering calpain processing.**

A Region details and sequence selection. Common residues to TrkB-FL and TrkB-T1 are in light blue and include a KFG sequence completely conserved in Trk proteins (filled black box). In the heatmap representation, dark red, white, and green represent, respectively, strong, weak, or no effect of point mutations, while black corresponds to wild-type residues. Black arrows denote theoretical calpain cleavage sites, while the red arrow points an experimentally established site. Sequences included in CPPs are emphasized by black rectangles. TM, transmembrane; TK, tyrosine kinase.

B CPP design. Peptides contain Tat aa 47–57 (italic) followed by the indicated rat TrkB-FL (green) or c-Myc (black) sequences.

C Confirmation of peptide entry into neurons. Cultures were incubated with Bio-TMyc (25 μM, 1 h) (image b) or left untreated (image a). Arrowheads highlight peptide permeability, detected by Fluorescein Avidin D (green), into neurons labeled with neuronal-specific antibody NeuN (red). Peptide is not detected in some neurons (arrow) and non-neuronal cells (asterisk). Confocal microscopy images correspond to single sections and are representative of five independent experiments. Scale bar, 10 μm.

D Effect of TMyc on neuronal survival. Primary cultures were incubated with TMyc (5, 15, or 25 μM, 30 min) and subjected to treatment with NMDA (100 μM) and glycine (10 μM) for 4 h. Specific neuronal viability was established and expressed relative to values in cultures with no treatment. Means ± SEM are represented (n = 8), and statistical analysis was performed by ANOVA test followed by *post hoc* Tukey's HSD test.

## TFL$_{457}$ interferes TrkB-FL downregulation and prevents acute and chronic excitotoxicity

Once validated the feasibility of our CPP approach, we analyzed the ability of the designed peptides to prevent TrkB-FL processing in excitotoxicity. Cultures were incubated with peptides TMyc, TFL$_{457}$, TFL$_{482}$, or TFL$_{541}$ before chronic NMDA treatment. Analysis of TrkB levels with an antibody recognizing all isoforms (panTrkB) showed that NMDA induced a significant decrease in TrkB-FL (145 kDa) and a reciprocal increase in truncated forms (TrkB-T1 and calpain-truncated TrkB-FL, collectively named tTrkB; Fig 2A), as described (Vidaurre *et al*, 2012). In cultures pretreated with TMyc, TFL$_{482}$, or

TFL$_{541}$, a similar TrkB-FL reduction was observed. Only TFL$_{457}$ interfered TrkB-FL downregulation (Fig 2B): Normalized receptor levels reached a value of 80 ± 7% compared with the drastic reduction observed in control cultures (40 ± 4%) or those treated with other CPPs. Calpain activation in excitotoxicity was confirmed by accumulation of characteristic spectrin breakdown products (BDPs; 150 and 145 kDa). Neuron-specific enolase (NSE), not affected by NMDA, was used as a loading control. Next, we established the effect of these peptides on neuronal viability (Fig 2C). No significant differences were found in cultures preincubated with TFL$_{482}$ or TFL$_{541}$ compared with cells treated with TMyc, which showed a marked time-dependent decrease in neuronal viability. In contrast,

TFL$_{457}$ increased neuronal viability at all evaluated times. Thus, in cultures preincubated with TFL$_{457}$, viability reached values of 50 ± 5% at 2 h of NMDA treatment, significantly higher than those obtained with TMyc (17 ± 2%). Likewise, we explored peptide TFL$_{518}$ and found that it was not able to maintain TrkB-FL levels and was not neuroprotective (Fig 2D and E). Altogether, we find a very good correlation between interference of TrkB-FL processing and neuronal death prevention, actions exerted by TFL$_{457}$ but absent for the other peptides.

Finally, we investigated TFL$_{457}$ neuroprotective effect in a different experimental paradigm where acutely induced damage precedes treatment. After brief excitotoxic stimulation, cells were treated with the NMDAR antagonist DL-AP5, together with TMyc or TFL$_{457}$, and neuronal viability was established 20 h later (Fig 2F). Excitotoxicity is an irreversible process that, after a critical period of time, cannot be blocked or reverted by NMDAR antagonists (Hartley & Choi, 1989). Hence, neuronal viability was reduced to 37 ± 9% by acute excitotoxicity in the presence of TMyc. However, damage was significantly lower in neurons treated with TFL$_{457}$ (71 ± 10), proving that TFL$_{457}$ is effective on neurodegenerative processes downstream NMDAR overactivation. Altogether, above results demonstrate that TFL$_{457}$ is neuroprotective either when present before a chronic excitotoxic stimulus or administered after acute damage.

### TrkB-FL preserved by TFL$_{457}$ action maintains Y816 phosphorylation and PLCγ-dependent pathways required for neuroprotection

BDNF binding induces transphosphorylation of TrkB-FL residues Y515 and Y816, outside the TK region, acting as anchor sites for adaptor proteins (Reichardt, 2006; Fig 3A). We verified the increase in TrkB phosphorylation induced by BDNF (1 h) with phosphospecific antibodies (pY515 and pY816; Appendix Fig S1A, upper panels). Notably, in our cellular model of excitotoxicity, pretreatment with BDNF for 1 h could not prevent the decrease in neuronal viability (Appendix Fig S1B) or interfere the processing of TrkB-FL (Appendix Fig S1C) induced by NMDA. Basal TrkB activation, observed in this experiment (Appendix Fig S1A, lower panels) as before (Du *et al*, 2003; Gomes *et al*, 2012; Vidaurre *et al*, 2012), might be due to spontaneous neuronal activity of dissociated cultures (Pasquale *et al*, 2017) and mediated by calcium-influx

through NMDARs and calcium channels (Du *et al*, 2003). Next, we were interested in establishing the phosphorylation status of the TrkB-FL receptor preserved in excitotoxic conditions by TFL$_{457}$ action. Compared to basal levels of TrkB phosphorylation in the absence of NMDA, an important decline was induced by the NMDAR agonist in TMyc presence (Fig 3B and C). In contrast, a more sustained Y816 phosphorylation was observed with TFL$_{457}$, both after direct lysate analysis (78 ± 9% vs. 46 ± 9) or following immunoprecipitation with a phosphoTyr-specific antibody (Fig 3D). No significant TFL$_{457}$ effect was found in pY515 levels (Fig 3C), despite TrkB-FL preservation.

To complete characterization of the downstream signaling pathways involved in TFL$_{457}$ neuroprotection, cultures were first preincubated with selective inhibitors of PI3K, MAPK/ERK, or PLCγ activities (Fig 3A), followed by peptide and NMDA treatment. Neuronal viability of cultures exposed to Wortmannin or UO126 and then treated with TFL$_{457}$ was similar to cells with no inhibitor (Fig 3E). Only PLCγ inhibition with U-73122 blocked TFL$_{457}$ effects, viability reaching values significantly lower than those obtained in cultures pretreated with TFL$_{457}$ but no inhibitor (27 ± 5% vs. 45 ± 4%) and similar to those found with TMyc and U-73122. Thus, the receptor preserved in excitotoxicity by TFL$_{457}$ action maintains Y816 phosphorylation and PLCγ-dependent signaling which is required for the neuroprotective effects. In contrast, pathways associated with Y515 phosphorylation are not involved in TFL$_{457}$ neuroprotection.

### TFL$_{457}$ preserves additional survival proteins downstream TrkB signaling in excitotoxicity

PLCγ activation increases the activity of $Ca^{2+}$-dependent pathways (Numakawa *et al*, 2001) which, among other effects, activate CREB. Thus, we investigated CREB activity in cultures preincubated with different concentrations of TMyc or TFL$_{457}$ (Fig 4A). Levels of total CREB or S133 phosphorylated CREB (pCREB), generally considered the active protein, showed a decrease induced by NMDA in the presence of TMyc. Reduction of pCREB is likely due to phosphatase activation, which causes CREB shut-off, blockade of BDNF expression, and death of mature neurons in excitotoxicity (Hardingham *et al*, 2002). Calpain processing might also contribute to CREB decrease, as suggested by CREB truncation in AD brains (Jin *et al*, 2013). Interestingly, levels of CREB and pCREB were preserved by TFL$_{457}$

---

**Figure 2.** **TFL$_{457}$ interferes TrkB-FL downregulation and prevents acute and chronic excitotoxicity.**

A   Effect of TFL$_{457}$, TFL$_{482}$, and TFL$_{541}$ on TrkB-FL levels. Cultures preincubated with TrkB-FL peptides or TMyc (25 µM, 30 min), followed by chronic NMDA treatment (2 h), were compared to cells without peptide. Levels of full-length (FL) TrkB were established with panTrkB, an antibody for the extracellular domain that also detects the truncated forms (tTrkB).

B   Quantitation of peptide interference of TrkB-FL downregulation. Receptor levels were normalized to NSE and expressed relative to values obtained in cultures without peptide or NMDA. Means ± SEM are represented, and analysis was performed by ANOVA test followed by *post hoc* Tukey's HSD test (*P = 0.02; n = 4).

C   Effect of TFL$_{457}$, TFL$_{482}$, and TFL$_{541}$ on neuronal viability after chronic excitotoxicity. Cultures were preincubated with peptides and treated with NMDA for 0–6 h as before. Values for each time point were represented relative to those of neurons incubated with the same peptide but no NMDA. Mean ± SEM of eleven (TFL$_{457}$), three (TFL$_{482}$), and seven (TFL$_{541}$) independent experiments is given. Data were analyzed by a two-way ANOVA test followed by *post hoc* Bonferroni test comparing values for each time point (***P = 0.0002, ***P = 0.00009, and *P = 0.0357, respectively for 2, 4, or 6 h).

D   Effect of TFL$_{518}$ on TrkB-FL levels. Cultures incubated with TFL$_{518}$ as before were compared to those treated with TMyc or TFL$_{457}$.

E   TFL$_{518}$ effect on neuronal viability. Cultures were treated and data analyzed as indicated in (C). No significant differences were found for TFL$_{518}$ while neuroprotection by TFL$_{457}$ was confirmed (***P = 0.0002 and **P = 0.002, respectively, for 2 or 4 h compared to TMyc; n = 3).

F   Effect of TFL$_{457}$ on neuronal viability after acute excitotoxicity. Cells were induced with NMDA (50 µM) and glycine (10 µM) for 1 h and, after agonists removal, culture proceeded for 20 h with DL-AP5 (200 µM) and TMyc or TFL$_{457}$ (15 µM). Mean ± SEM relative to cultures incubated with TMyc but no NMDA is represented. Analysis was performed by unpaired Student's *t*-test (*P = 0.04; n = 5).

Source data are available online for this figure.

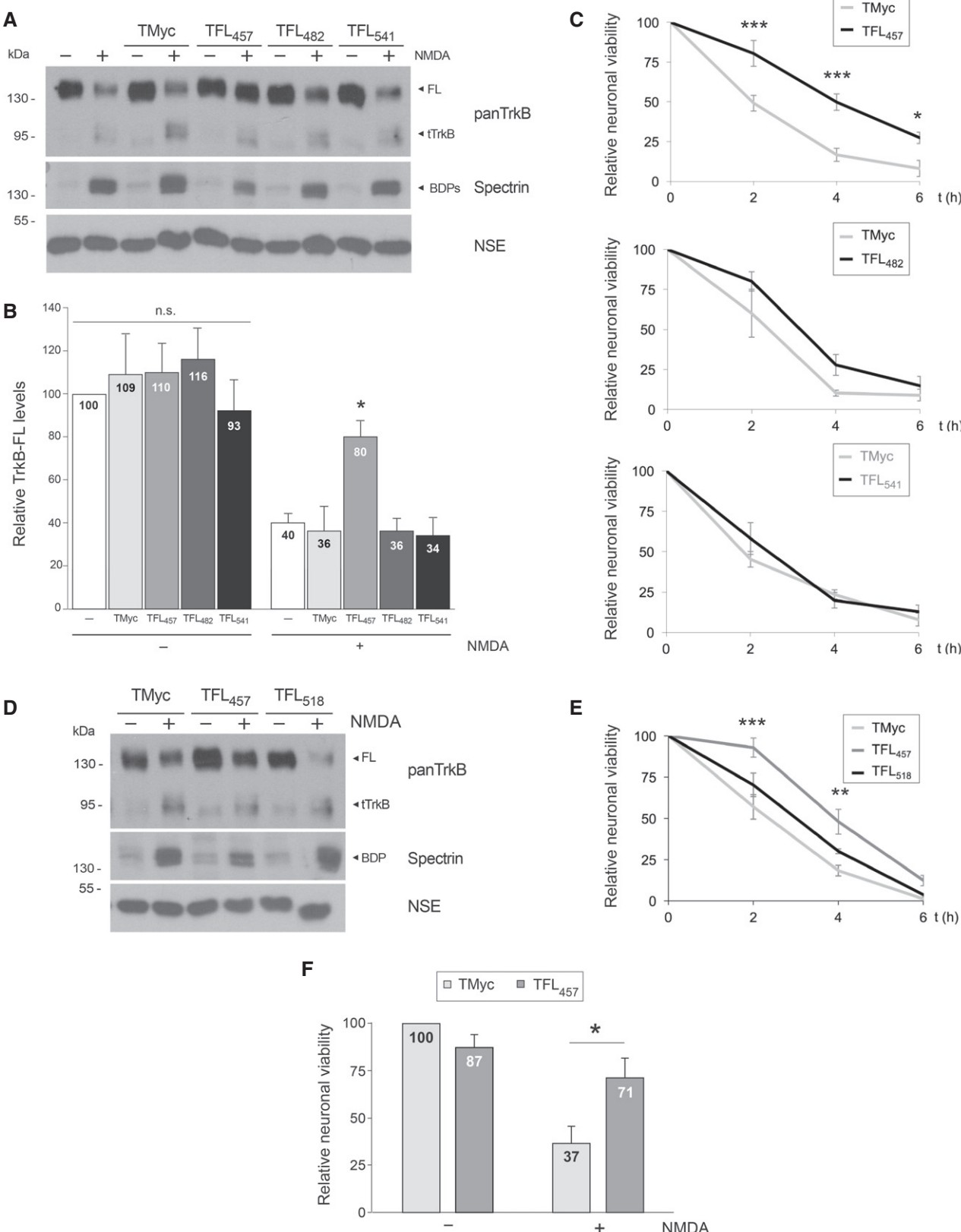

**Figure 2.**

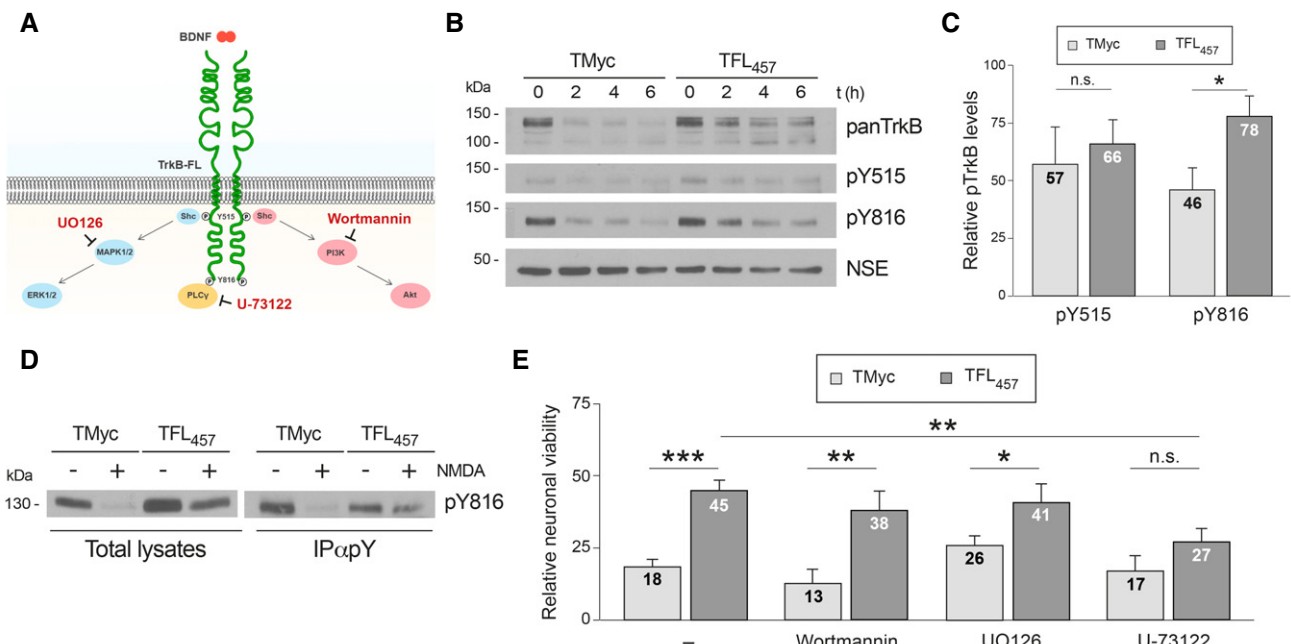

**Figure 3. TrkB-FL preserved by TFL$_{457}$ action maintains Y816 phosphorylation and PLCγ-dependent pathways required for neuroprotection.**

A   TrkB-FL main signaling cascades with indication of anchoring residues, tested drugs, and their targets.

B   Effect of TFL$_{457}$ on Y515 and Y816 phosphorylation. Cultures preincubated with TMyc or TFL$_{457}$ (25 μM, 30 min) were treated with NMDA (0–6 h) and analyzed with panTrkB or phosphospecific antibodies.

C   Quantitation of pY515 and pY816. Mean ± SEM of normalized pY515 ($n = 7$) or pY816 ($n = 4$) levels obtained after NMDA treatment (2 h) is represented relative to those found in cells with the same peptide but no NMDA. Statistical analysis was performed by unpaired Student's $t$-test (*$P = 0.046$; n.s. = non-significant).

D   Analysis by immunoprecipitation of TFL$_{457}$ effect on pY816 levels. Cultures preincubated and treated as before with NMDA (2 h) were immunoprecipitated with antibodies specific for phosphorylated tyrosine (pY). pY816 was analyzed by WB in immunoprecipitates.

E   TFL$_{457}$ effects on neuronal viability after inhibition of TrkB-FL signaling. Cultures were preincubated (30 min) with inhibitors specific for PI3K (Wortmannin, 100 nM), MAPK/ERK (UO126, 300 nM), or PLCγ (U-73122, 5 μM) before incubation with TMyc or TFL$_{457}$ (25 μM, 30 min). Viability was established 4 h after NMDA treatment. Means ± SEM relative to untreated cultures are represented, and data were analyzed by ANOVA test followed by *post hoc* Tukey's HSD test (***$P = 0.0001$, **$P = 0.009$, *$P = 0.02$, n.s. = non-significant, respectively, for TMyc vs. TFL$_{457}$ in untreated or Wortmannin, UO126, or U-73122-treated cells; **$P = 0.008$ for untreated vs. U-73122-treated cells preincubated with TFL$_{457}$; $n = 5$).

Source data are available online for this figure.

in a dose-dependent way, parallel to peptide effects on neuronal viability (Appendix Fig S2).

We further characterized TFL$_{457}$ effects on survival pathways by time-course experiments (Fig 4B and C). As expected, a restrained TrkB-FL reduction was found at 2 and 4 h of NMDA treatment in the presence of TFL$_{457}$ compared with TMyc. The effect was isoform-specific, and TFL$_{457}$ did not prevent the gradual increase in TrkB-T1 characteristic of *in vitro* excitotoxicity (Vidaurre *et al*, 2012). Next, we analyzed TFL$_{457}$ effects on prosurvival CREB and MEF2D proteins. The latter belongs to a family of TFs involved in neuronal survival promotion downstream BDNF/TrkB activation (Liu *et al*, 2003) or synaptic activity (Linseman *et al*, 2003). Similarly to CREB, MEF2 targets include *Bdnf* and, thus, there might be a feedback mechanism between BDNF expression and TF activation. A severe decrease in MEF2D was found in cultures subjected to excitotoxicity in TMyc presence, probably due to action of caspases or calpain (Tang *et al*, 2005; Wei *et al*, 2012), while this TF was better preserved by TFL$_{457}$ (Fig 4C). As before, TFL$_{457}$ had important effects in CREB and pCREB levels.

The intricate interplay of glutamate and neurotrophin-signaling made us wonder if TFL$_{457}$ might also maintain NMDAR-dependent survival pathways. Excitatory neurotransmission stimulates BDNF

synthesis (Hardingham *et al*, 2002) which modulates NMDAR activity through TrkB activation (Bamji *et al*, 2006). Additionally, both signaling cascades activate CREB and MEF2. We analyzed NMDAR obligatory subunit GluN1 and GluN2A, predominantly confined to synapses of mature neurons and mostly related to survival (Papadia & Hardingham, 2007). As expected, NMDA induced subunit downregulation in cells treated with TMyc (Gascon *et al*, 2005, 2008) which was interfered by TFL$_{457}$ (Fig 4B). Finally, analysis of spectrin processing established the specificity of TFL$_{457}$ effects. Together, these results demonstrate that, by interfering TrkB-FL downregulation, TFL$_{457}$ is not only able to partly preserve BDNF/TrkB signaling but also interconnected NMDAR survival pathways that cooperate to sustain CREB and MEF2D levels. These prosurvival TFs might then regulate expression of members of both cascades and, thus, amplify TFL$_{457}$ effects on survival.

## Promoter activity and mRNA levels of CREB and MEF2-regulated genes are preserved by TFL$_{457}$ in excitotoxicity

First, we explored if CREB and MEF2 promoter activities were maintained in excitotoxicity by TFL$_{457}$ action. Transcription dependent on these TFs is key in neuronal survival induced by neurotrophins

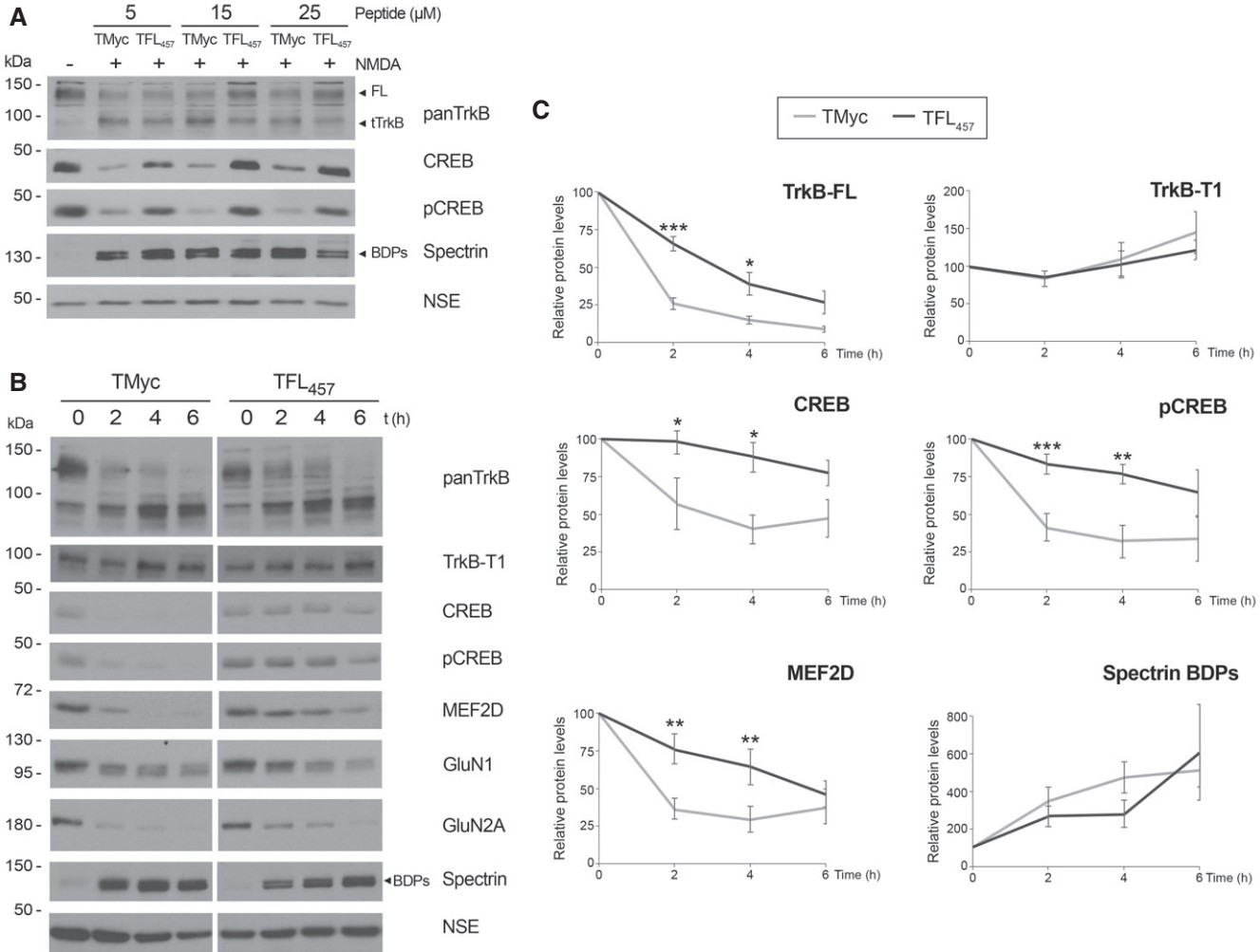

**Figure 4. TFL$_{457}$ preserves additional survival proteins downstream TrkB signaling in excitotoxicity.**

A   Dose-dependent TFL$_{457}$ effects. Cultures incubated with TMyc o TFL$_{457}$ (5, 15, or 25 μM, 30 min) were treated with NMDA (2 h) and compared to cells without peptide or NMDA.

B   Time-course TFL$_{457}$ effects. Cells preincubated with peptides (25 μM, 30 min) were treated with NMDA (0–6 h).

C   Quantitation of peptide effects. Normalized protein levels are presented relative to values obtained in cells incubated with the same peptide but no NMDA.
    Means ± SEM are given. Results were analyzed by a two-way ANOVA test followed by *post hoc* Bonferroni test, comparing TMyc- or TFL$_{457}$-treated cells for each time point (TrkB-FL, ***$P$ = 0.0009 and *$P$ = 0.02; CREB, *$P$ = 0.0149 and *$P$ = 0.0148; pCREB, ***$P$ = 0.0007 and **$P$ = 0.0012; MEF2D, **$P$ = 0.0023 and **$P$ = 0.0047. All values are, respectively, for 2 or 4 h; $n$ = 6).

Source data are available online for this figure.

(Bonni *et al*, 1999; Liu *et al*, 2003) or synaptic activity (Lonze & Ginty, 2002; Linseman *et al*, 2003). We performed reporter assays using promoters with minimal MEF2 or CREB responsive elements (respectively, pMEF2 or pCRE; Woronicz *et al*, 1995; Deogracias *et al*, 2004) controlling luciferase expression. In pMEF2-transfected cells (Fig 5A), luciferase activity was 90 ± 12% after NMDA treatment in the presence of TFL$_{457}$, a value significantly higher than that obtained with TMyc (58 ± 8%). This effect was specific and lost by mutation of MEF2 responsive elements. In pCRE-transfected cells (Fig 5B), excitotoxicity decreased luciferase activity to 36 ± 5 and 64 ± 6%, respectively, in TMyc or TFL$_{457}$-treated cells, proving a TFL$_{457}$ effect also on CREB-dependent regulation. Treatment with KG-501, a compound that disrupts interactions of CREB-binding

protein (CBP) with different TFs that include CREB, decreased all luciferase values that now were similar in excitotoxic cultures treated with TFL$_{457}$ or TMyc. Similar reporter assays were performed with promoter regions corresponding to CREB and/or MEF2-regulated genes expressing prosurvival proteins GluN1 (Bai *et al*, 2003) and GluN2A (Desai *et al*, 2002), BDNF (Tao *et al*, 1998), and TrkB (Deogracias *et al*, 2004). The decrease in GluN1 and GluN2A promoter activities induced by excitotoxicity was again counteracted by preincubation with TFL$_{457}$ compared with TMyc (Fig 5C). For BDNF and TrkB, a significant TFL$_{457}$ effect could not be observed with the severe excitotoxic conditions used above, but a strong effect was unveiled under milder excitotoxicity (Fig 5D).

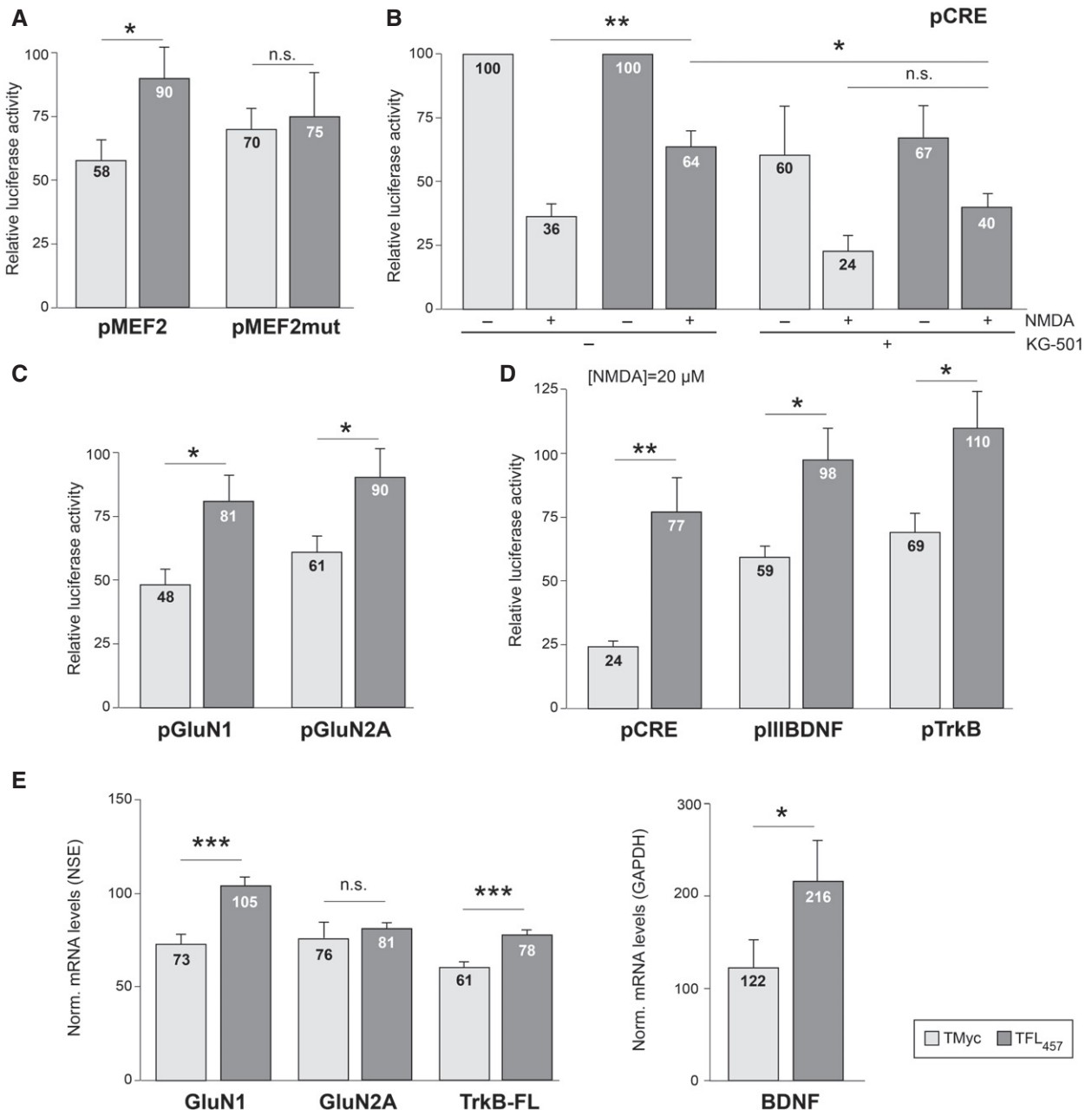

**Figure 5. Promoter activity and mRNA levels of CREB and MEF2-regulated genes are preserved by TFL₄₅₇ in excitotoxicity.**

A   Effect of TFL₄₅₇ on MEF2-promoter activity. Cultures transfected with pMEF2 (two minimal wild-type MEF2 elements) or pMEF2mut (mutant) were preincubated with CPPs (25 μM, 30 min) and treated with NMDA (100 μM, 2 h). Means ± SEM of luciferase activities obtained in excitotoxicity, relative to values found in cells treated with same peptide and no NMDA, are presented. Significance was analyzed by Student's *t*-test (*P = 0.039; n.s. = non-significant, P = 0.79; n = 8).

B   Effect on CRE-promoter activity. pCRE contains two minimal CREs. Peptide preincubation was as above with or without KG-501 (10 μM). Mean ± SEM luciferase activities is given relative to values in cells treated with the same peptide but no NMDA or KG-501. Differences found in excitotoxicity were analyzed by ANOVA test followed by *post hoc* Tukey's HSD test (*P = 0.023; **P = 0.003; n.s. = non-significant, P = 0.20; n = 6).

C   Effect on NMDAR-subunit promoters. Cells transfected with pGluN1 or pGluN2A were treated and analyzed as in panel (A). Significance was analyzed by Student's *t*-test (pGluN1, *P = 0.016; pGluN2A, *P = 0.029; n = 14).

D   Effects on BDNF promoter III or TrkB promoter. Cells transfected with pIIIBDNF (n = 9), pTrkB (n = 7), or pCRE (n = 5), as a control, were processed as above but using 20 μM NMDA for 4 h. Data are presented and analyzed as in panel (A). Significance was analyzed by Student's *t*-test (pCRE, **P = 0.0098; pIIIBDNF, *P = 0.029; pTrkB, *P = 0.033).

E   Effects of TFL₄₅₇ on mRNA levels of CREB/MEF2-regulated genes. Total RNA was extracted from cultures preincubated with CPPs (25 μM, 30 min) and treated or not with NMDA (100 μM, 4 h). Levels of mRNA were normalized to NSE (genes expressed in neurons, left panel) or GAPDH (neuronal and glial expression, right panel). Means ± SEM of levels obtained in excitotoxicity relative to values found in cells treated with same peptide and no NMDA are presented. Differences found in excitotoxicity were analyzed by ANOVA test followed by *post hoc* Tukey's HSD test (GluN1, ***P = 0.00096; GluN2A, P = 0.98; TrkB-FL, ***P = 0.00037; BDNF, *P = 0.045; n = 8).

These results in promoter activities suggested that, for genes regulated by these TFs, $TFL_{457}$ might be able to tune transcriptional changes induced by excitotoxicity and favor neuroprotection. In accordance with previous results (Gascon *et al*, 2005), we observed a decrease in levels of *Grin1* mRNA, encoding GluN1, in cells treated with TMyc and NMDA (Fig 5E). This decay was reverted by $TFL_{457}$ action. For *Ntrk2*, we could also verify the decrease in the mRNA encoding the TrkB-FL isoform induced by NMDA in the presence of TMyc (Vidaurre *et al*, 2012), changes again counteracted by $TFL_{457}$. A similar tendency was found for GluN2A regulation. Finally, BDNF mRNA increased in excitotoxic conditions in the presence of TMyc according to published results (Zafra *et al*, 1991), accumulation exacerbated by $TFL_{457}$. In conclusion, $TFL_{457}$ not only preserves CREB and MEF2 proteins in excitotoxicity but also their regulatory activities which very likely modify transcription of target genes central to neuronal survival, contributing in this way to neuroprotection.

## Preservation of TrkB-FL in the cell surface by $TFL_{457}$ precedes interference of RIP and calpain processing in excitotoxicity

We investigated next the primary mechanism of $TFL_{457}$ action on TrkB-FL stability. Receptor processing by calpain or RIP produces characteristic intracellular fragments that can be recognized by C-ter antibodies (TrkB-FL Ct; Tejeda *et al*, 2016). Remarkably, interference of TrkB-FL cleavage by $TFL_{457}$ in excitotoxicity was associated with a significant decrease in both RIP and calpain C-ter fragments, respectively, f42/39 (42–39 kDa) and f32 (32 kDa; Fig 6A and B). We hypothesized that $TFL_{457}$ might be interfering an excitotoxicity-induced process upstream of TrkB-FL processing and neuronal death. Therefore, the combination of $TFL_{457}$ with specific calpain (Fig 6C) or metalloproteinase (Fig 6D) inhibitors would not have additive neuroprotective effects. For TMyc-treated cultures, we found a modest but significant effect of calpain inhibitors on neuronal viability ($31 \pm 2\%$ vs. $19 \pm 2\%$), according to previous observations (Rami *et al*, 1997; Gerencser *et al*, 2009; Wei *et al*, 2012). This small effect is probably related to the opposite roles played by the major brain calpain isoforms regarding neuroprotection and neurodegeneration (Baudry & Bi, 2016). Nevertheless, the effect of $TFL_{457}$ was very similarly independent of calpain inhibition. Likewise, the neuroprotective effect of $TFL_{457}$ was not significantly affected by GM6001, a broad metalloproteinase inhibitor blocking the first step of RIP (Fig 6D). Thus, we suggest that the actions of $TFL_{457}$ on neuronal viability occur upstream of TrkB-FL processing.

Interestingly, $TFL_{457}$ could counteract actions of both proteolytic systems on TrkB-FL although the peptide contained receptor sequences away from the experimentally established calpain site (Jeronimo-Santos *et al*, 2015) and putative metalloproteinase targets. As mentioned, the TrkB-FL sequences included in $TFL_{457}$ are inside an IDR located in a juxtamembrane region involved in regulation of receptor recycling through protein interaction (Huang *et al*, 2009, 2011; Zamani *et al*, 2018). Thus, we hypothesized that $TFL_{457}$ effects on TrkB-FL stability might be indirectly mediated by a switch in recycling that places the receptor in a subcellular location where it is less susceptible to proteolysis. Cultures briefly treated with NMDA (1 h), to limit excitotoxicity-induced degradation, showed a strong decrease in cell-surface TrkB-FL levels which was similar to that described for BDNF (Fig 6E; Zheng *et al*, 2008). The

ratio of surface to total TrkB-FL levels, which subtracts the effect of partial receptor processing, was not statistically different for NMDA- or BDNF-treated cultures. Next, we tested if $TFL_{457}$ might be able to interfere the decrease in TrkB-FL cell-surface expression induced by excitotoxicity (Fig 6F). At this early time of NMDA treatment, the reduction in total receptor levels was not significantly different in TMyc or $TFL_{457}$-treated cultures as expected. In contrast, surface expression of TrkB-FL was preserved after treatment with NMDA in the presence of $TFL_{457}$ ($98 \pm 22\%$), differently from results obtained TMyc-treated cultures where NMDA induced a dramatic decrease ($17 \pm 12\%$). Therefore, $TFL_{457}$ enhances TrkB-FL localization inside the plasma membrane after NMDAR overactivation.

Finally, we investigated the importance of TrkB-FL endocytosis for receptor proteolysis in excitotoxicity. We used a mouse model of permanent ischemia where the excitotoxic process occurs *in vivo* and causes profound cortex neurodegeneration as established by Fluoro-Jade C staining (Fig EV3). Human stroke is frequently caused by embolic or thrombotic occlusion of small arteries, a situation mimicked in this model by microvascular photothrombosis (Pevsner *et al*, 2001). Mice were injected with dynasore, to inhibit dynamin-dependent endocytosis, or vehicle before damage induction in motor and somatosensory areas (Fig 6G). Animals presented emergent cortical infarcts only 3 h after insult initiation, proving that damage is early developed in this ischemia model (Appendix Fig S3). At this early time of damage, levels of TrkB-FL and TrkB-T1 were decreased in the infarcted area (I) of vehicle treated animals compared with the contralateral region (C) (Fig 6H and I), as described (Tejeda *et al*, 2016). However, TrkB-FL levels were significantly higher in the infarcted area of animals pretreated with dynasore compared with control animals ($69 \pm 21\%$ vs. $19 \pm 4\%$). In contrast, dynasore had no significant effect on previously described TrkB-T1 downregulation. Thus, we conclude that *in vivo* prevention of endocytic processes induced by ischemia results in TrkB-FL stabilization. Altogether, experiments above support that a primary mechanism of $TFL_{457}$ action might be the interference of TrkB-FL endocytosis and/or enhancement of receptor recycling, resulting in a global increase in TrkB-FL levels in the membrane where the receptor might be less susceptible to proteolytic processing.

## $TFL_{457}$ counteracts TrkB-FL downregulation in ischemia and reduces infarct volume and neurological damage

Our next goal was to investigate if $TFL_{457}$ could also interfere TrkB-FL downregulation *in vivo* and, therefore, act as an ischemia neuroprotectant. Preliminary experiments using a biotinylated CPP derived from a previously described neuroprotective peptide (Bio-NA-1; Aarts *et al*, 2002) showed correct delivery of Tat peptides to undamaged mice cortex (Fig EV4). We also demonstrated that biotinylated $TFL_{457}$ (Bio-$TFL_{457}$) was able to reach cortical neurons in the brain area that will be damaged in our ischemia model (Fig 7A). The *in vivo* analysis of $TFL_{457}$ neuroprotective potential required an improvement of peptide stability in plasma. One approach is the modification of the N-ter and C-ter sequences by, respectively, acetylation and amidation to mimic the natural protein structure. In this way, we generated MTMyc and MT$TFL_{457}$ (Appendix Table S1). As seen before for $TFL_{457}$, MT$TFL_{457}$ was able to interfere TrkB-FL processing *in vitro* (Appendix Fig S4) and also

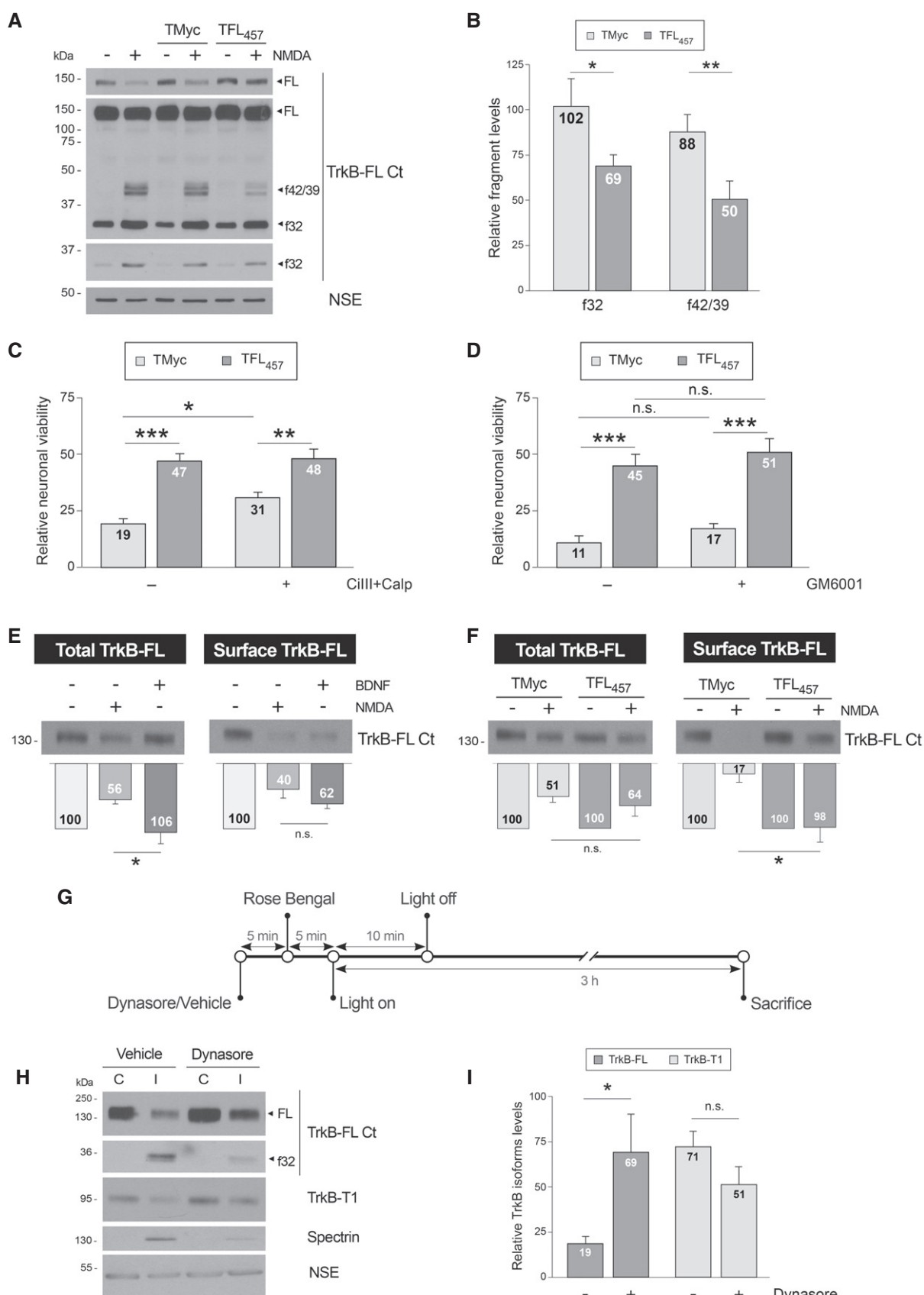

Figure 6.

**Figure 6.  Preservation of TrkB-FL in the cell surface by TFL$_{457}$ precedes interference of RIP and calpain processing in excitotoxicity.**

A   TFL$_{457}$ interference of RIP and calpain processing in excitotoxicity. Cultures were preincubated with peptides (25 μM, 30 min) or left untreated before NMDA addition (2 h). Different exposures are shown to facilitate visualization of FL, f42/39, or f32.

B   Quantitation of normalized f32 and f42/39 levels. Means ± SEM are shown relative to cultures treated with NMDA and no peptide. Analysis was performed by unpaired Student's $t$-test (*$P$ = 0.014, **$P$ = 0.00198; $n$ = 6).

C   Effect of TFL$_{457}$ on neuronal viability after calpain inhibition. After 30 min preincubation with calpeptin (Calp, 10 μM) and calpain inhibitor III (CiIII, 10 μM), cultures were treated with CPPs and NMDA (4 h) as before. Means ± SEM relative to those obtained in untreated cultures are given. We performed ANOVA test followed by post hoc Tukey's HSD test (*$P$ = 0.035, **$P$ = 0.0058, ***$P$ = 0.00009; $n$ = 12).

D   Effect of TFL$_{457}$ on neuronal viability after metalloproteinase inhibition. Cultures preincubated with GM6001 (10 μM, 30 min) were treated with CPPs and NMDA as before. Analysis was performed as above (without GM6001, ***$P$ = 0.0007; with GM6001, ***$P$ = 0.0006; $n$ = 4).

E   Effect of NMDA on total and cell-surface TrkB-FL levels. Cultures were incubated with NMDA (100 μM) or BDNF (100 ng/ml) for 1 h. Membrane proteins were labeled and purified and compared to corresponding total lysates. TrkB-FL levels are expressed relative to untreated cells. Mean ± SEM is presented and analyzed by Student's $t$-test (*$P$ = 0.027; n.s. = non-significant, $P$ = 0.1; $n$ = 7).

F   Effect of TFL$_{457}$ on the decrease in TrkB-FL surface levels induced by NMDA. Analysis was performed as before in cells incubated with CPPs (25 μM, 30 min) and NMDA-treated (1 h). Results obtained in excitotoxic cultures are expressed relative to cells treated with the same peptide but no NMDA. Data are analyzed as in panel (E). Mean ± SEM is presented and analyzed by Student's $t$-test (*$P$ = 0.019; n.s. = non-significant, $P$ = 0.44; $n$ = 4).

G   Experimental design to study the effect of endocytosis inhibition on TrkB-FL downregulation in ischemia. Mice were i.p. injected with vehicle or dynasore followed by Rose Bengal. Vessel occlusion and brain damage were induced by cold-light irradiation. Animals were sacrificed early after damage initiation.

H   Effect of endocytosis inhibition on TrkB-FL downregulation. The infarcted area of the ipsilateral hemisphere (I) was compared to the corresponding region of the contralateral area (C). Results from representative mice injected with dynasore or vehicle are shown. Different exposures are presented to facilitate visualization of dynasore effects on TrkB-FL and f32.

I   Quantitation of TrkB-FL and TrkB-T1 in the infarcted area. Normalized protein levels are expressed relative to those of the corresponding contralateral region. Mean ± SEM is presented and analyzed by Student's $t$-test (*$P$ = 0.04; n.s. = non-significant, $P$ = 0.14; $n$ = 9).

Source data are available online for this figure.

had a significant effect in vivo (Fig 7C and D). For these protection assays, the CPPs were injected after damage initiation (Fig 7B) to mimic a clinical situation where drugs would be provided after insult. The strong decrease in TrkB-FL levels observed early after damage (3 h) in the infarcted area of animals injected with MTMyc (44 ± 4%) was restrained in those treated with MTFL$_{457}$ (75 ± 8%; Fig 7D). Next, we examined the long-term effects that early TrkB-FL stabilization by MTFL$_{457}$ might have on infarct volume. We performed TTC staining of coronal sections obtained 24 h after brain insult (Fig 7E), a time when infarcts are well established. Interestingly, infarct volume in MTFL$_{457}$-treated animals was 8 ± 0.5% of the total hemisphere volume and 11 ± 0.5% in the MTMyc group, which represents a 28% reduction after MTFL$_{457}$ treatment (Fig 7F). Moreover, this neuroprotective effect correlated with an improvement in balance and motor coordination in the beam-walking test. We found a decrease in 42% in the number of slips due to MTFL$_{457}$ treatment compared with MTMyc animals (6 ± 1 vs. 10 ± 2; Fig 7G).

In conclusion, these results demonstrate that peptide MTFL$_{457}$ is neuroprotective in vivo. In a severe model of permanent brain ischemia, it reduces the infarct size and neurological damage in parallel to preservation of TrkB-FL from excitotoxicity-induced downregulation. Additionally, our data unveil the importance of this neurotrophin receptor for neuronal survival after brain damage and lead the way for the development of rational stroke therapies that efficiently support neurotrophic signaling.

## Discussion

Herein, we present the design and characterization of a neuroprotective peptide able to interfere downregulation of BDNF receptor TrkB-FL induced by excitotoxicity. This effect correlates with increased viability of cortical neurons after NMDAR overactivation in vitro (acute or chronic) or in vivo (ischemic insult). Peptide TFL$_{457}$ has a short TrkB-FL juxtamembrane region (aa 457–471)

inner to an IDR, sequences involved in disease pathways now considered as therapeutic targets (Chen & Kriwacki, 2018). Also, this IDR is located in a receptor region (aa 453–536) previously defined as important for regulation of TrkB-FL endocytic recycling through Hrs binding (Huang et al, 2009). This region also interacts with other proteins such as suppressor of cytokine signaling 2 (SOCS2), which regulates TrkB trafficking (Zamani et al, 2018), c-Jun NH$_2$-terminal kinase-interacting protein 3 (JIP3) that mediates interaction with kinesin-light chain (KLC1; Huang et al, 2011) or ubiquitin C-terminal hydrolase L1 (UCH-L1; Guo et al, 2017).

A link between Trk trafficking and signaling is critical for neuronal functioning. Electric stimulation enhances the BDNF response of neurons by increasing cell-surface TrkB (Zhao et al, 2009). Ca$^{2+}$-influx also improves BDNF responsiveness by increasing TrkB internalization into signaling endosomes (Du et al, 2003). In contrast, little is known about how NMDAR overactivation affects the levels of TrkB at the cell surface. Endocytosis is enhanced in excitotoxicity by a clathrin/dynamin-mediated mechanism preceding neuronal death in vitro (Vaslin et al, 2007) or ischemia (Vaslin et al, 2009). Here, we demonstrate that a brief excitotoxic insult induces a strong cell surface decrease in TrkB-FL previous to processing. By competing protein interactions, TFL$_{457}$ might interfere excitotoxicity-induced receptor endocytosis or facilitate recycling back to the membrane. In fact, endocytosis inhibition strongly reduces TrkB-FL cleavage in ischemia, suggesting that receptor downregulation requires previous endocytosis. TFL$_{457}$ efficiently prevents receptor decrease at the surface of excitotoxic neurons, exposing aa 457–471 as relevant to TrkB-FL trafficking in excitotoxicity. This switch in localization would interfere TrkB-FL proteolysis, protect prosurvival pathways, and finally lead to a reduced neuronal death.

This hypothesis implies that TrkB-FL processing occurs mostly outside plasma membrane. The major mechanism of TrkB-FL downregulation in excitotoxicity is cleavage by calpain (Tejeda et al, 2016), ubiquitous protease central to neuronal death in stroke, and excitotoxicity-associated neurodegenerative diseases (Bevers &

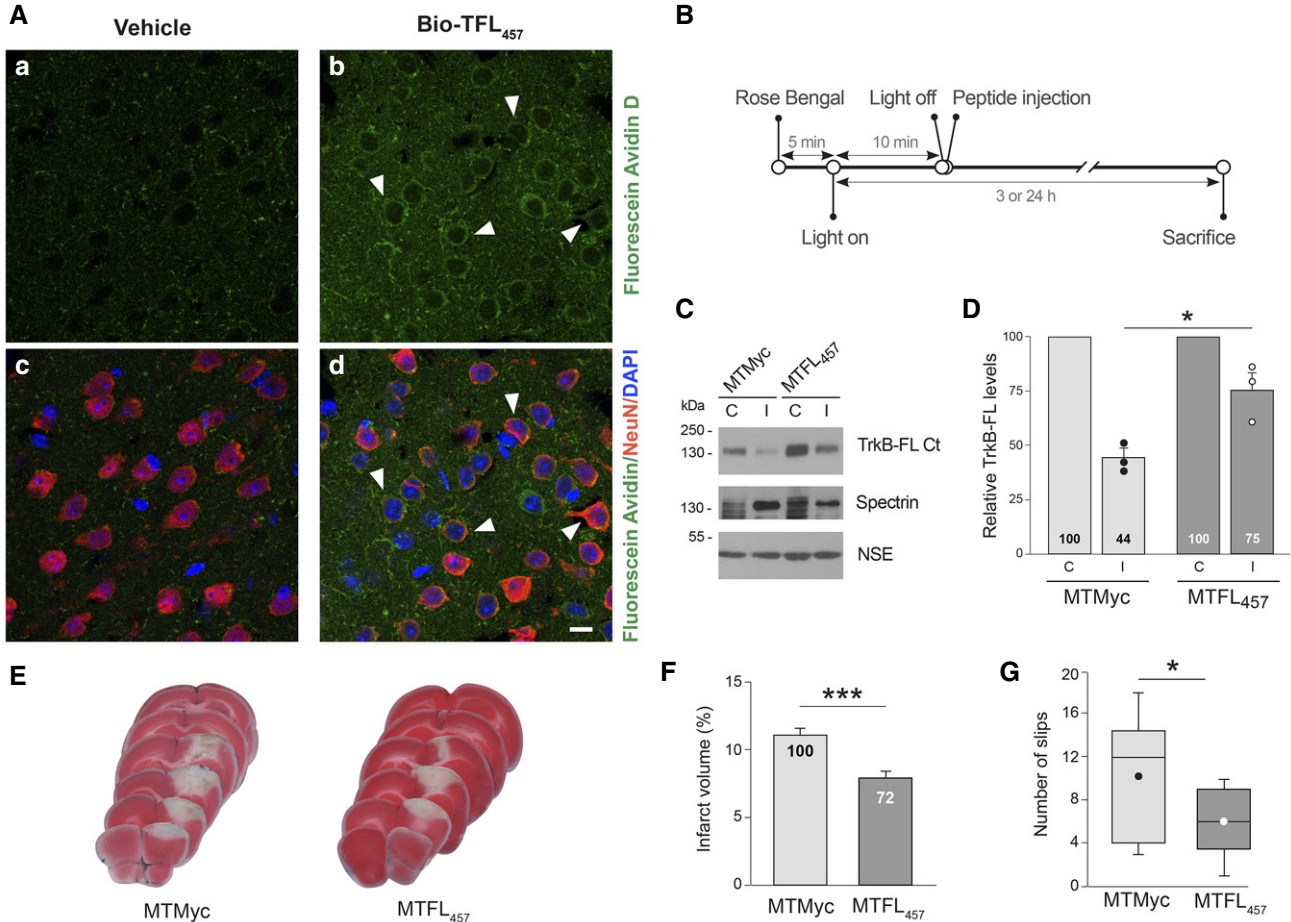

**Figure 7.  TFL₄₅₇ counteracts TrkB-FL downregulation in ischemia and reduces infarct volume and neurological damage.**

A   Confirmation of Bio-TFL₄₅₇ delivery to mice cortex. Biotinylated TFL₄₅₇ (Bio-TFL₄₅₇, 4 nmol/g, retro-orbitally injected) was detected in coronal sections stained with NeuN and DAPI. Representative confocal microscopy images of cortical areas correspond to single sections. Scale bar, 10 μm.

B   Timeline to analyze *in vivo* effects of MTMyc and MTFL₄₅₇. Mice subjected to ischemia as before were retro-orbitally injected with CPPs (10 nmol/g) 10 min after damage initiation.

C   Representative analysis of TrkB-FL at 3 h of damage. Comparison of the infarcted area (I) and the corresponding contralateral region (C) is shown.

D   Quantitation of TrkB-FL levels. Means ± SEM of normalized TrkB-FL levels in the infarcted area relative to the corresponding contralateral region are presented and results analyzed by Student's *t*-test (*$P$ = 0.021; $n$ = 3).

E   Representative 1 mm brain coronal slices stained with TTC after 24 h of insult corresponding to animals injected with MTMyc or MTFL₄₅₇.

F   Infarct volume of animals injected with MTMyc or MTFL₄₅₇ expressed as a percentage of the hemisphere volume. Means ± SEM are given. Differences were analyzed by Student's *t*-test (***$P$ = 0.00008; $n$ = 14). Results for MTFL₄₅₇ are also expressed as percentage of the infarct volume in animals injected with MTMyc.

G   Evaluation of balance and motor coordination. Number of contralateral hind paw slips were measured after 24 h of damage. Data spread is presented by box and whisker plots showing interquartile range, median, minimum, and maximum values. Means (dots) ± SEM were also calculated and analyzed by Student's *t*-test (*$P$ = 0.04; $n$ = 11).

Source data are available online for this figure.

Neumar, 2008). The subcellular localization of calpain processing is under debate. Activation is favored near plasma and endosomal membranes (Tompa *et al*, 2001). Thus, cleavage of vascular endothelial cadherin (VE-cadherin) occurs in clathrin-enriched membrane domains during endocytosis (Su & Kowalczyk, 2017). Activation can also occur in microdomains with high local [Ca²⁺] or other compartments, such as Golgi membranes, nucleus, or mitochondria (Bevers & Neumar, 2008). RIP, secondary mechanism of TrkB-FL downregulation (Tejeda *et al*, 2016), is regulated by intracellular trafficking of the proteases (Morohashi & Tomita, 2013). Accordingly, A disintegrin and metalloproteinase (ADAM) 10 and 12 surface location is regulated by endocytosis and can be altered

by disease (Stautz *et al*, 2012; Marcello *et al*, 2013). Activity of γ-secretase can also occur along endocytic and recycling pathways (Tarassishin *et al*, 2004). Further experiments will establish the precise location of TrkB processing.

The central role of calpain in cell physiology has steered strategies to block protease actions on specific substrates, avoiding generic effects (Wu & Tymianski, 2018). Thus, we have developed a neuroprotective CPP (Tat-K) that by interfering cleavage of Kinase D-interacting substrate of 220 kDa (Kidins220), a Trk and NMDAR-interacting protein, is able to preserve the activity of ERK1/2 and CREB survival pathways and the viability of neurons after *in vitro* excitotoxicity (Gamir-Morralla *et al*, 2015). In contrast, we could

not prevent TrkB-FL downregulation or neuronal death with $TFL_{518}$, although it contains an experimentally established calpain site (Jeronimo-Santos *et al*, 2015). It is interesting that TrkB-FL-interacting protein FRS2, which anchors at pY515 nearby the cleavage site, has low affinity for short peptides compared with longer sequences which probably can adopt proper three-dimensional conformations (Zeng *et al*, 2014). However, the design of peptides longer than $TFL_{518}$ might confront unwanted side effects by potential displacement of adaptor proteins and aberrant signaling.

The receptor preserved by $TFL_{457}$ in excitotoxicity is not completely functional due to reduced Y515 phosphorylation, probably caused by tyrosine phosphatase Shp-2 (Src homology-2 domain-containing phosphatase-2) activation. Shp-2 is involved in a $Ca^{2+}$-induced feedback mechanism suppressing further pY515-dependent TrkB-FL activation (Rusanescu *et al*, 2005). In contrast, $TFL_{457}$ preserves Y816 phosphorylation and promotes neuroprotection dependent on PLCγ, neurotrophin-stimulated enzyme involved in neuronal survival (Yamada *et al*, 2002). Most PLCγ effects are mediated by $IP_3$-promoted $Ca^{2+}$-release from internal stores (Numakawa *et al*, 2001) and activation of enzymes such as PKC or $Ca^{2+}$-calmodulin-regulated protein kinases (Huang & Reichardt, 2003). We propose that PLCγ contributes to $TFL_{457}$ neuroprotection through activation of prosurvival TFs CREB and MEF2. The importance of PLCγ for CREB activation is well established (Minichiello *et al*, 2002). Antidepressants increase TrkB-dependent CREB phosphorylation in neurons through PLCγ-signaling with no effect on Y515 phosphorylation (Rantamaki *et al*, 2007). Here, we observe that $TFL_{457}$ preserves in excitotoxicity the promoter activity of several genes containing CRE and MEF2 elements. The genes encoding CREB and MEF2 TFs are probably also affected by this peptide since their own promoters contain, respectively, CRE (Zhang *et al*, 2005) and MEF2 elements (Cripps *et al*, 2004). Interestingly, $TFL_{457}$ has also an impact on the mRNA levels of prosurvival genes regulated by these TFs (*Bdnf, Ntrk2, Grin1, and Grin2a*) counteracting the effects of excitotoxicity. This suggests that this peptide starts a positive autoregulatory mechanism that highly potentiates its neuroprotective efficacy at early times and might be also critical for long-term neuronal regeneration.

$TFL_{457}$ might be highly relevant to human stroke therapy since it efficiently prevents TrkB-FL downregulation, and reduces infarct size and neurological damage in a severe model of permanent ischemia. The developed CPP provides a promising alternative/complement to the use of BDNF to recover aberrant BDNF/TrkB signaling pathways after brain damage. Although this neurotrophin has been considered for the treatment of neurological and psychiatric disorders, clinical trials with BDNF have been unsuccessful (Tejeda & Diaz-Guerra, 2017). One reason might be that the type of neuronal death induced by NMDAR overactivation (Ankarcrona *et al*, 1995) determines the cell response to BDNF. Using neocortical cultures (neurons and glial cells), we have confirmed previous observations demonstrating that BDNF fails to protect neurons from excitotoxicity produced by necrotic mechanisms (Koh *et al*, 1995). In contrast, BDNF could attenuate apoptotic neuronal death induced in these same cultures by different stimulus (Gawg *et al*, 1997) or in an alternative cellular paradigm of excitotoxicity (semi-pure hippocampal neurons; Gomes *et al*, 2012; Lau *et al*, 2015) where NMDA activates apoptosis (Almeida *et al*, 2005). Since both types of neuronal death occur in the ischemic brain, the *in vivo* efficacy of

BDNF administration might be limited to a subpopulation of the degenerating neurons. Another challenge to the use of BDNF in brain therapy is aberrant function of TrkB-FL receptor, already described in stroke and other disorders, which might be prevented by neuroprotective peptide $TFL_{457}$.

Until now, most CPP-based strategies for stroke have targeted signaling cascades related to NMDARs (Wu & Tymianski, 2018). Recently, a Tat peptide containing GluN2B C-ter (NA-1; Aarts *et al*, 2002) has provided the first strong evidence that neuroprotection is feasible in human ischemic stroke (Hill *et al*, 2012). NA-1 dissociates GluN2B-PSD-95-nNOS complexes, uncoupling NMDARs from nNOS neurotoxic signaling, and is currently under Phase 3 clinical trials (FRONTIER and ESCAPE-NA-1). Other NMDAR-targeted CPPs under development disrupt p53 interaction with death-associated protein kinase 1 (DAPK1; Pei *et al*, 2014), prevent nuclear translocation of PTEN (phosphatase and tensin homolog deleted on chromosome 10; Zhang *et al*, 2013), or selectively reduce mGluR1 calpain processing (Xu *et al*, 2007). The novelty of the approach presented here is that $TFL_{457}$ targets a different signaling cascade, likewise critical to neuronal survival: the BDNF/TrkB pathway. By affecting TrkB-FL trafficking, $TFL_{457}$ blocks receptor processing and allows sustained survival signaling. Remarkably, $TFL_{457}$ has additional effects on NMDAR-dependent cascades and, therefore, has a double action on these major survival pathways, converging in CREB activation. Thus, $TFL_{457}$ might also improve endogenous neurogenesis (Lonze & Ginty, 2002) and promote functional recovery of ischemic brain. The versatility of $TFL_{457}$ could be further increased by combination with other neuroprotective CPPs, an approach suggested by the complexity of the ischemic process. Thus, if proven efficient also *in vivo*, peptide Tat-K (Gamir-Morralla *et al*, 2015) could be used simultaneously with $TFL_{457}$ to block defective neurotrophic support and NMDAR signaling at different levels. As mentioned before, another interesting possibility would be the combined action of $TFL_{457}$ with BDNF mimetic or strategies to increase BDNF availability (Tejeda & Diaz-Guerra, 2017). In summary, the stabilization of receptor TrkB-FL by the neuroprotective peptide $TFL_{457}$ has a great potential to recover TrkB and NMDAR-dependent neuroprotective pathways not only in acute stroke but also in other neurological conditions associated with excitotoxicity.

## Materials and Methods

All reagents, materials, mouse models, and software used in the study are described in Appendix Table S1.

### Experimental models

All animal procedures were performed in compliance with European Union Directive 2010/63/EU and were approved by the CSIC and Comunidad de Madrid (Ref PROEX 221/14) ethics committees. The housing facilities at the Institute were approved by Comunidad de Madrid (# ES 280790000188) and comply with official regulations. Animals had a standard health and immune status and were looked after by professional caretakers, being checked daily. Male mice were kept in groups of up to five in standard IVC cages while 1–2 pregnant rats occupied standard cages, all of them containing bedding and nesting material. They were under controlled lighting

conditions (12 h light cycles), relative humidity and temperature, irradiated food and water provided *ad libitum*. All efforts were made to minimize animal suffering and reduce the number of animals sacrificed.

## Mice model of ischemia by photothrombosis

Permanent focal ischemia was induced in the cerebral cortex of adult male Balb/cOlaHsd mice (25–30 g; 8–12 weeks of age; Harlan Laboratories, Boxmeer, The Netherlands) by microvascular photothrombosis. The model mimics embolic or thrombotic occlusion of small arteries, frequently found in human stroke, and causes a focal brain damage with histological and MRI correlations to human patterns (Pevsner *et al*, 2001). Upon receipt, animals were allowed to acclimatize to our facilities for at least 1 week prior to ischemic induction, kept with the same cage mates, and daily inspected to check normal health status. Mice were anesthetized with isoflurane (5% for induction, 2% for maintenance in oxygen; Abbot Laboratories, Madrid, Spain) and then placed in a stereotaxic frame (Narishige Group, Tokyo, Japan). Body temperature was maintained at 36–37°C using a self-regulating heating blanket (Cibertec, Madrid, Spain). A midline scalp incision was made, the skull was exposed, and both Bregma and Lambda points were identified. A cold-light (Schott KL 2500 LCD; Schott Glass, Mainz, Germany) with a fiber optic bundle of 1.5 mm in diameter was centered using a micromanipulator on the right side, at 0.2 mm anterior and 2.0 mm lateral (+0.2 AP, +2 ML) relative to Bregma. Afterward, the photosensitive dye Rose Bengal (7.5 mg/ml, prepared in sterile saline; Sigma-Aldrich) was administered by retro-orbital injection of the venous sinus, for intravenous (i.v.) vascular access, to a body dose of 20 mg/kg. Five minutes later, the brain was illuminated (600 lms, 3,000 K) through the intact skull for 10 min. Development of brain injury involves damage to the vascular endothelium, platelet activation, and subsequent microvascular thrombotic occlusion of the irradiated region (Watson *et al*, 1985). According to the Paxinos mouse brain atlas, the areas underneath this stereotaxic position that result irradiated are the primary motor cortex and the primary somatosensory cortex (hindlimb and forelimb).

As indicated, a single dose of dynasore (30 mg/kg) or vehicle (20% DMSO, 10% Tween-20 in saline) was injected intraperitoneally 5 min before Rose Bengal administration. For neuroprotection, we retro-orbitally injected a single dose (10 nmol/g) of peptides MTMyc or MTFL$_{457}$ (> 95% purity; GenScript) 10 min after damage initiation, right after irradiation completion. These peptides are N-ter acetylated and C-ter amidated versions of TMyc or TFL$_{457}$ (Appendix Table S1), aimed to improve plasma stability. They were solubilized as 2.5 mM solutions in 0.9% NaCl, and, just before injection, fresh HCO$_3$NH$_4$ was added to a final concentration of 44 mM to neutralize the acidity derived from trifluoroacetic acid present in them. Mice were not subjected to other procedures before ischemia and were naïve to drug or peptide treatment.

After completion of the surgical procedure, the incision was sutured and mice were allowed to recover. For immunoblot analysis, animals were sacrificed 3 h after damage induction by CO$_2$ inhalation followed by cervical dislocation and their brains were sectioned into serial 2-mm-thick coronal slices using a mouse brain matrix (Stoelting, Wood Dale, IL, USA). Slices were briefly stained

with a cold 2% solution of triphenyltetrazolium chloride (TTC, Sigma-Aldrich) to avoid endogenous postmortem calpain activation. The unstained area of the cerebral cortex in the ipsilateral hemisphere (defined as infarcted tissue, I), as well as the corresponding region in the contralateral hemisphere (C), was dissected to prepare protein lysates. For assessment of infarct volume, animals were sacrificed as before 24 h after damage induction and their brains were sectioned into serial 1-mm-thick coronal slices using a mouse brain matrix as before. Slices were then completely stained with 2% TTC at room temperature and fixed in 4% paraformaldehyde before scanning of both rostral and caudal sides.

## Primary culture of rat cortical neurons

Primary neuronal cultures were prepared from the cerebral cortex of 18-day-old Wistar rat embryos (E18), both genders being indistinctly used. Dissected cerebral cortices were mechanically dissociated in culture medium (Minimum Essential Medium, Life Technologies, cat#21090-022) supplemented with 22.2 mM glucose, 0.1 mM glutamax, 5% fetal bovine serum, 5% donor horse serum, 100 U/ml penicillin, and 100 µg/ml streptomycin similarly as described before (Choi, 1985; Choi *et al*, 1987). The cell suspension was seeded at a density of $1 \times 10^6$ cells/ml in the same medium using plates previously treated with poly-L-lysine (100 µg/ml, Sigma-Aldrich) and laminin (4 µg/ml, Sigma-Aldrich) overnight at 37°C. Unless otherwise indicated, glial growth was inhibited after 7 days DIVs by adding cytosine β-D-arabinofuranoside (AraC, 10 µM) and experimental treatments took place after 13 DIVs.

## Induction of neuronal excitotoxicity

To induce chronic excitotoxicity, cultures were incubated with NMDA (100 µM, unless otherwise stated) and its co-agonist glycine (10 µM), a treatment herein denoted simply as NMDA. The co-agonists induce a strong excitotoxic response in the mature neurons present in the culture but have no effect on astrocyte viability (Choi, 1985; Choi *et al*, 1987). When indicated, primary cultures were preincubated for 30 min with Tat-derived CPPs (> 95% purity; GenScript; 25 µM, unless otherwise indicated) or BDNF (100 ng/ml) before NMDA treatment. Peptides or BDNF were kept in the medium along treatment. For protease inhibition, cultures were preincubated with the indicated compounds (10 µM CiIII, 10 µM calpeptin, or 10 µM GM6001) for 30 min before peptide preincubation and NMDA addition, the inhibitor remaining in the medium for the duration of treatment. Enzymatic activities downstream TrkB activation were inhibited with selective inhibitors Wortmannin (100 nM), UO126 (300 nM), or U-73122 (5 µM), blocking, respectively, PI3K, MAPK/ERK, or PLCγ, added to culture media 30 min before peptide preincubation and induction of excitotoxicity as before. KG-501 (10 µM), an inhibitor of the interactions established by the co-activator CBP with different transcription factors, including CREB, was used similarly.

To induce acute excitotoxicity, primary cultures were treated with NMDA (50 µM) and glycine (10 µM) for 1 h. Cells were then washed and fed with conditioned medium without NMDAR agonists but containing the generic antagonist DL-AP5 (200 µM), together with peptides TMyc or TFL$_{457}$ (15 µM). The effect of acute NMDA treatment was analyzed 20 h later.

                                                       

## Assessment of neuronal injury in cultures

We used the MTT reduction assay to measure cell viability. At the end of treatments, MTT (0.5 mg/ml, Sigma-Aldrich) was added to the medium and, after 2 h of incubation at 37°C, the formazan salts formed were solubilized in DMSO and spectrophotometrically quantified at 570 nm. As primary cultures contain neurons and glial cells, we established the contribution of glia viability to total values by exposing sister cultures to 400 μM NMDA, 10 μM glycine for 24 h before MTT assay. These conditions induce nearly complete neuronal death and no glial damage. After subtracting this absorbance value, we obtained the viability of the neuronal subpopulation. All experiments included sample triplicates for each treatment, and multiple independent experiments were carried out as detailed in the figure legends. Generally, the viability of neurons preincubated with a peptide and subjected to excitotoxicity was calculated relative to that of neurons preincubated with the same peptide but no NMDA. However, for some experiments, neuronal viability was expressed relative to the untreated cultures or those preincubated with control peptide TMyc but no NMDA.

## Western blot (WB) analysis

Cultures and brain tissue were lysed in RIPA buffer (50 mM Tris–HCl pH 8, 150 mM NaCl, 1% sodium deoxycholate, 1% NP-40, 1 mM DTT, and 0.1% SDS for cultures or 1% SDS for tissue) containing protease and phosphatase inhibitors (complete protease and PhosSTOP phosphatases inhibitor cocktail tablets, Roche). Protein concentration was established with BCA Protein Assay Kit (Thermo Fisher). Total cell lysates were denatured in SDS-sample buffer followed by heating at 95° for 5 min. Equal amounts of total cell lysates were resolved in Tris-Glycine SDS–PAGE and transferred on to a nitrocellulose membrane (GE Healthcare). Membranes were stained with a Ponceau S solution to check for protein transference efficacy. After blocking with a 5% nonfat dry milk solution in Tris-buffered saline (TBS) with 0.05% Tween-20, membranes were incubated overnight at 4°C with primary antibodies and then washed and incubated with appropriate anti-rabbit or anti-mouse peroxidase-conjugated secondary antibodies (Santa Cruz Biotechnology or Bethyl) for 1 h at room temperature. Finally, immunoreactivity was visualized using Clarity Western ECL Blotting Substrate (BioRad) and band intensity was quantified by densitometric analysis (ImageJ). The levels of the protein of interest were normalized using those of neuron-specific enolase (NSE) present in the same sample and expressed relative to values obtained in their respective controls, arbitrarily given a 100% value. NSE was used as a neuronal loading control since it is not affected by NMDA treatment. In contrast, the activation of calpain induced by excitotoxicity was confirmed by analyzing the formation of characteristic breakdown products (BDPs; 150 and 145 kDa) from spectrin, a standard substrate of this protease. Multiple independent experiments were carried out and quantitated as detailed in the figure legends.

## Immunoprecipitation

Total cell lysates were prepared from cortical cultures in 1% NP-40, 80 mM NaCl, 20 mM EDTA, and 20 mM Tris–HCl (pH 8), containing protease and phosphatase inhibitors as before. Cleared lysates (1 mg) were precipitated overnight at 4°C with 4 μg of an anti-phosphotyrosine-specific antibody (Appendix Table S1) before addition of 60 μl of 50% Protein G Agarose, incubated for 1 h at room temperature with agitation. Equal amounts of lysates or equivalent volumes of immunoprecipitated complexes were analyzed by WB as indicated.

## Biotinylation of cell-surface proteins

After treatment, cultures were immediately washed with ice-cold PBS containing 1 mM CaCl₂ and 0.5 mM MgCl₂. Surface proteins were biotinylated for 30 min at 4°C with 0.5 mg/ml of NHS-SS-biotin (Thermo Fisher) prepared in the same buffer. Free biotin excess was eliminated washing cultures with cold PBS containing 1 mM CaCl₂, 0.5 mM MgCl₂, and 0.1% BSA, followed by two additional washes omitting BSA. Lysis was in RIPA buffer with protease and phosphatase inhibitors but without DTT. Except for a total TrkB-FL aliquot, extracts were incubated with streptavidin resin (GenScript) for 3 h at 4°C to precipitate biotinylated proteins. After washing the streptavidin–biotin complexes twice with lysis buffer containing 500 mM NaCl and protease and phosphatase inhibitors, plus two additional washing steps omitting the NaCl, pellets were solubilized and denature in SDS–PAGE sample buffer (10 min at 50°C). Equivalent volumes of isolated proteins and total extracts were analyzed by WB in parallel. Independent experiments were carried out and quantified as detailed in figure legends.

## Peptide visualization in primary cultures

Cultures grown on coverslips (DIV 13) were incubated for 1 h with biotin-conjugated TMyc (Bio-TMyc, 25 μM, GenScript, Appendix Table S1) or left untreated. Cells were fixed with 4% paraformaldehyde in PBS (30 min) and blocked and permeabilized with 1% BSA and 0.1% Triton X-100 in PBS (30 min). Coverslips were incubated with anti-NeuN (3 h) followed by secondary antibodies conjugated to Alexa Fluor 546, Fluorescein Avidin D (50 μg/ml), and DAPI (5 μg/ml). Mounting was in Prolong Diamond. Confocal images were acquired using an inverted Zeiss LSM 710 laser confocal microscope (Jena, Germany) with a 40× (quantitation) or 63× Plan-Apochromatic oil immersion objective (details of entry) and were normalized for each color separately. Images respectively corresponded to maximum intensity projections or single sections and were processed for presentation with ImageJ (NIH Image). A minimum of 100 neurons were counted in five independent experiments. Mean ± SEM is given.

## Peptide visualization in brain cortex

Mice retro-orbitally injected with vehicle (saline), biotin-labeled NA-1 (Bio-NA-1), or TFL₄₅₇ (Bio-TFL₄₅₇; Appendix Table S1), used at 4 nmol/g, were deeply anesthetized 30 min after peptide administration and intracardially perfused with cold PBS and 4% paraformaldehyde in PBS. Brains were post-fixed in the same fixative at 4°C for 24 h and cryoprotected in 30% sucrose for 48 h at 4°C. Coronal frozen sections (30 μm thick) obtained using a cryostat (Leica, Heidelberg, Germany) were incubated in blocking solution (10% goat serum, 0.5% Triton X-100 in PBS) for 3 h at room temperature, followed by anti-NeuN in 4% goat serum (1:500) for 3 h. After washing, sections were incubated 1 h with Alexa Fluor

546-conjugated antibodies (1:500), Fluorescein Avidin D (200 μg/ml), and DAPI (5 μg/ml) prepared as before. Sections were mounted and dried on slides, and cover slipped with Prolong Diamond. Tile image acquisition was performed using an inverted laser confocal microscope as before with a Zeiss LD LCI Plan-Apochromat 25×/0.8 Imm Corr DIC M27 objective (1.2× zoom). Other images were obtained with a 40× objective. Images were processed as described and correspond to single sections. Background was subtracted using vehicle-injected animals.

### Fluoro-Jade C staining

The infarcted tissue present in the cerebral cortex of mice sacrificed 5 h after photothrombosis was first identified by Nissl staining of brain coronal sections. Specific labeling of degenerating neurons and cell nuclei was then performed in adjacent sections by Fluoro-Jade C staining. To that, sections were immersed in a basic alcohol solution (1% NaOH in 80% ethanol) for 5 min and rinsed for 2 min in 70% ethanol and for two additional min in distilled water. Next, they were incubated in a 0.06% potassium permanganate solution for 10 min, rinsed for 2 min in distilled water, and transferred for 10 min to a fresh solution of 1 μg/ml Fluoro-Jade C (Millipore) in 0.1% acetic acid containing 0.5 μg/ml DAPI. Sections were then washed in distilled water, mounted on slides, allowed to dry overnight at room temperature, and cover slipped with Prolong Diamond. Fluorescent images were acquired using an Eclipse 90i Nikon microscope and a DS-Qi1Mc Nikon digital camera. Pictures were processed with NIS-Elements 3.001, ImageJ (NIH Image), and Adobe Photoshop CC software (Adobe Systems Inc.).

### Cell transfection and gene reporter assays

Neurons were transfected with indicated plasmids (Appendix Table S1) in neurobasal medium using Lipofectamine 2000 according to manufacturer instructions. The plasmids contained different gene promoter sequences upstream of the firefly luciferase reporter gene. After 2 h, DNA–liposomes complexes were removed and neurons fed with conditioned medium collected before. Treatments with drugs and peptides were started after completing 24 h of transfection. Lysis was in *Passive Lysis Buffer* (Promega, Cat# E1941), and luciferase activity was determined in 25 mM glycylglycine, 15 mM $SO_4Mg$, 4 mM EGTA, 15 mM potassium phosphate (pH 7.8), 3.3 mM ATP, 1 mM DTT, and 75 μM luciferin. Independent experiments, including quadruplicated samples, were repeated a minimum of five times as indicated.

### RNA extraction and real-time PCR analysis

Total RNA extracted using QIAcube technology was treated with DNases before cDNA synthesis using a "High Capacity cDNA Reverse Transcription Kit" (Applied Biosystems). A 7900HT Fast real-time PCR system (Applied Biosystems) was used for SYBR green gene expression assays with indicated primer sequences (Appendix Table S1). A specific standard curve was performed in parallel for each gene, and technical triplicates were prepared for each sample. PCR conditions were 10 min at 95°C, followed by 40 cycles of 15 s at 95°C and 60 s at 60°C. Data from eight independent experiments were analyzed and normalized to levels of housekeeping genes NSE (genes of neuronal expression) or GAPDH (genes expressed in neurons and glial cells).

### Measurement of infarct volume

Rostral and caudal images of TTC-stained coronal slices were analyzed using ImageJ software by an observer blinded to experimental groups. After image calibration, delineated areas of ipsilateral and contralateral hemispheres, and the infarcted region (unstained area) were measured. Considering slices thickness, the corresponding volumes were calculated and corrected for edema's effect, estimated by comparing total volumes of hemispheres. The corrected infarct volumes were expressed as percentage relative to the contralateral hemisphere, to correct for normal size differences between different animals. For each animal, the mean of results obtained for rostral and caudal sides was calculated.

### Beam-walking test

Motor coordination and balance were evaluated in mice right before and 24 h after the ischemic insult by measuring the number of contralateral hind paw slips in the beam walk apparatus. Mice had to walk through a narrow beam (1 m × 1 cm × 1 cm) placed 50 cm above the tabletop, going from an aversive stimulus (60-W light bulb) to a black goal box with nesting material. Slips taking place in a previously selected central beam segment (50 cm long) were counted. Before damage induction, mice were allowed to cross the beam once, to get acquainted with the test, which they repeated 24 h after photothrombosis, immediately before sacrifice.

### Quantification and statistical analysis

All data were expressed as mean ± standard error of the mean (SEM) of at least three independent experiments. The details of the number of experiments done, the precise sample size, and the specific statistical test applied for each case can be found in the figure legends. For *in vivo* studies, animals were randomly allocated to the experimental groups and the researchers doing the experiments were blind respect to treatment. In order to reduce subjective bias, we used a numerical key for blinded measurement of infarct size or behavioral assessment. There was not a previous estimation of sample size, and experiments were independently carried out as indicated. We pre-established an exclusion criteria for values more than three standard deviations outside the mean, considered as outliers. For the *in vitro* studies, cell samples in each independent experiment were sister primary cultures grown in multiwell plates (whenever possible), always obtained from the same suspension of cortical cells. Treatments were assigned in a random way, and biological replicates ($n = 3$ or $n = 4$) were included for most experiments as indicated. Multiple independent experiments were then carried out as detailed in the figure legends. Statistical significance was determined by unpaired Student's *t*-test, one-way ANOVA followed by Tukey's HSD *post hoc* test, or two-way ANOVA followed by *post hoc* Bonferroni. Data were represented as percent of controls or maximum values as indicated. The box and whisker plot displays the data spread and five-number summary (minimum, first quartile, median, third quartile, and maximum) together with the mean value for each set of data. A *P*-value smaller than 0.05

### The paper explained

#### Problem

Stroke is a great health and social problem worldwide. It is the second leading cause of death, the major cause of adult disability, and the second of dementia. Pharmacological therapies for ischemic stroke (85% of cases) are still limited to thrombolytic drugs, which can be only administered to very few patients.

#### Results

Herein, we have developed a neuroprotective compound (peptide $TFL_{457}$) directed to reduce the death of those neurons surrounding the infarct core, which initially survive stroke but die in the following hours/days. The main mechanism of this neuronal death is a process known as excitotoxicity that, among other things, inhibits the functioning of a pathway critical for neuronal survival due to the breakdown of receptor TrkB-FL. By preventing TrkB-FL cleavage, $TFL_{457}$ triggers a positive auto-regulatory mechanism that highly potentiates its neuroprotective efficacy and increases the levels of several proteins important for survival of neurons. This neuroprotective peptide could be highly relevant for stroke therapy since, in a mouse model of ischemia, it similarly prevents TrkB-FL downregulation, efficiently decreases the infarct size, and improves balance and motor coordination in a neurological test.

#### Impact

This neuroprotective peptide might be efficient in minimizing the effects of stroke but also other acute or chronic neurological disorders which are also associated with excitotoxicity, such as several neurodegenerative diseases.

was considered statistically significant ($*P < 0.05$, $**P < 0.01$, $***P < 0.001$). Statistical analysis was performed with the Statistical Package for Social Science (SPSS, v.19, IBM). To select the correct statistical test, data were analyzed before by SPSS for outliers, normal distribution (Kolmogorov–Smirnov and Shapiro–Wilk's tests), and homogeneity of variances (Levene's test).

**Expanded View** for this article is available online.

## Acknowledgements

We acknowledge funding from Ministerio de Economía y Competitividad (BFU2013-43808-R and BFU2016-75973-R). The cost of publication has been paid in part by FEDER funds. Contracts were funded associated with projects BFU2013-43808-R (G.S.T and G.M.E-O) and BFU2016-75973-R (G.M.E-O and E.SA). We generously received plasmids from Dr. Winoto (Universidad de California; pRSRF and pRSRFmut), Dr. Rodriguez-Peña (IIB-CSIC, Spain; pCRE and pTrkB), Dr. Bai (Universidad de Maryland; pNRL 5.4 Kb), and Dr Buonano (NICHD, NIH; pGluN2). We are grateful to Drs. Lorrio, Sobrado, and Ayuso-Dolado for technical advice with the ischemia model, Drs. Belinchón and Guerrero for microscopy assistance, and Dr. Giralt for help with peptides. We also thank Drs. Iglesias and Rodriguez-Peña for helpful discussions.

## Author contributions

Conceptualization, MD-G and GST; Methodology, MD-G, GST, and GME-O; Formal Analysis, MD-G, GST, ESA, OGV, and GME-O; Investigation, GST, GME-O, OGV, and ESA; Writing—Original Draft, MD-G; Writing—Review & Editing, MD-G, GST, and GME-O; Visualization, MD-G and GST; Supervision, MD-G; Funding Acquisition, MD-G.

## Conflict of interest

The authors declare that they have no conflict of interest.

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
