## [Review Process File · EMBO Molecular Medicine]

Prevention of excitotoxicity-induced processing of BDNF receptor TrkB-FL leads to stroke neuroprotection

Gonzalo S. Tejada, Gema M. Esteban-Ortega, Esther San Antonio, Óscar G. Vidaurre and Margarita Díaz-Guerra

Review timeline:

Submission date:	15 October 2018
Editorial Decision:	9 November 2018
Revision received:	11 March 2019
Editorial Decision:	4 April 2019
Revision received:	9 April 2019
Editorial Decision:	29 April 2019
Revision received:	2 May 2019
Accepted:	3 May 2019

Editor: Céline Carret

Transaction Report:

1st Editorial Decision

9 November 2018

Thank you for the submission of your manuscript to EMBO Molecular Medicine. We have now heard back from the two referees whom we asked to evaluate your manuscript.

You will see that while referee 1 is mainly concerned about technical issues, hampering a proper evaluation of your work, referee 2 is more supportive and provides interesting suggestions to further improve and develop your findings. Upon our cross-commenting exercise, referee 1 would like to see more convincing *in vitro* data and a better define link to the *in vivo* findings. I would like to encourage you to develop your study as suggested by referees, provide numerical data regarding transfection efficiencies, detailed methods, accurate literature, better discussions and experimental evidence to support the findings as suggested by ref.2.

We would welcome the submission of a revised version within three months for further consideration and would like to encourage you to address all the criticisms raised as suggested to improve conclusiveness and clarity. Please note that EMBO Molecular Medicine strongly supports a single round of revision and that, as acceptance or rejection of the manuscript will depend on another round of review, your responses should be as complete as possible.

I look forward to receiving your revised manuscript.

***** Reviewer's comments *****

Referee #1 (Comments on Novelty/Model System for Author):

Unless the efficiency of transfection method used can be drastically improved the authors should rely on alternative methods such as the use of protease inhibitors affecting all cultured cells or else use model systems with all cultured cells expressing TrkB mutations thought to be targeted by proteases.

Referee #1 (Remarks for Author):

The study by Tejada et al. examines an important and interesting issue related to TrkB signalling, and especially signalling duration. Exposing TrkB to its ligands including BDNF in particular has long been known to cause a rapid loss of cell surface, full-length TrkB. Much of the initial work in this area focussed on down-regulation as a result of ligand-mediated receptor internalisation. However, recent work by several groups begun to concentrate on additional mechanisms including proteolysis, especially in the context of glutamate-mediated excitotoxicity. The new twist in the story is that increased calcium influx through the NMDA receptor may activate calpain(s) and preventing calpain activity has already been shown to significantly reduce TrkB proteolysis (see in particular Danelon et al MCN 2016, not quoted). Also, a recent study by Jeronimo-Santos actually mapped a calpain cleavage site on TrkB and indicated that various protease inhibitors reduce Aβ induced proteolysis of TrkB. For reasons that are not entirely clear given this context, the authors of the present study explored additional potential calpain cleavage sites on TrkB based on computer-based predictions. They then went on to design several peptides based on these predictions and coupled these to a cell permeable TAT sequence. These reagents were then used to explore a possible reduction of cytotoxicity exerted by NMDA on cultures neurons. The most promising peptide is also used in a test of local cortex infarction caused by photothrombosis. While the topic is of considerable interest and so is the translational potential there are significant issues with the study.

1. The most worrisome point is the use of cell-permeable peptides. As Fig. 1C illustrates only a fraction of the neurons is actually transfected. This fraction appears to be quite small. Also related to this important validation Figure, it is not clear what the arrows in the DAPI panel actually points to as nuclei are not clearly visible. Given the additional problem that the proportion of transfected cells may vary from experiment to experiment, the transfection strategy at the core of the study does not appear to be suitable for biochemical analyses such as those performed in the study. Obviously, the key assumption underlying mass analysis of cultured cells is that all or nearly all neurons are transfected and this condition is not fulfilled here. An additional problem is that according to Fig. 1C out of only three transfected cells one seem to be a non-neuronal cell.

2. Given the emphasis on calpain throughout it is difficult to understand why the calpain inhibitor used (Fig. 6) does not seem to be particularly effective, and apparently less so than a peptide designed on the basis of the assumption that calpain cleavage is at the core of TrkB proteolysis caused by exposure to NMDA. This also contrasts with the results of previous studies.

3. The authors need to more explicitly acknowledge including in the Introduction that using a direct biochemical approach, a previous study clearly mapped including amino-terminal sequencing of a calpain fragment generated by cleavage of recombinant TrkB. Whilst this previous work is mentioned in Fig 1 (red arrow) and in the Discussion it is surprising that the authors chose to first describe their guess work about potential cleavage sites based on prediction programmes before acknowledging highly relevant, convincing previous work.

4. Given the uncertainties related to the in vitro work aiming at validating the use of a particular peptide, the in vivo data are difficult to interpret, not least those related to dynamin-mediated endocytosis.

Referee #2 (Remarks for Author):

Ischemic stroke produces an infarct core surrounding an area called penumbra where neurons may subsequently die of excitotoxicity induced by overstimulation of the NMDA receptor. In the penumbra, BDNF expression is increased, but TrkB levels are reduced due to excitotoxicity-induced degradation. The authors hypothesize that preservation of TrkB in the penumbra would reduce the damage of stroke to the brain. They designed several cell-penetrating peptides (CPPs) containing TrkB intracellular sequences surrounding potential calpain cleavage sites. One CPP containing the TrkB juxtamembrane sequence (TFL457) was found to reduce TrkB degradation, preserve TrkB-PLCgamma1 signaling and increase neuronal viability in neuronal cultures after excitotoxicity. The authors further showed that the preservation effect of the peptide was due to increased levels of surface TrkB. Therefore, peptide TFL457 reduces TrkB endocytosis or enhances TrkB recycling back to the plasma membrane by sequestering proteins that normally interact with the TrkB juxtamembrane domain. Importantly, the authors showed that administration of peptide TFL457 reduced infarct size in a mouse ischemic model. These results are interesting, given that very few neuroprotective strategies are available to treat stroke. A few technical issues should be addressed before the paper is published.

1. According to the authors' previous work, degradation of full-length TrkB should increase the abundance of tTrkB. This expected result is not consistently shown in Figure 1A.
2. In Figure 5, the authors employed luciferase constructs containing a minimal promoter to show peptide TFL457 increases transcription of GluN1, GluN2A, BDNF and TrkB. Because the constructs do not have all regulatory elements, luciferase expression may not reflect the expression of endogenous genes. The results should be confirmed using real-time RT-PCR to measure levels of mRNAs from these genes.
3. The authors used NA-1 peptide to show that the peptide is able to get into the brain. Why did not the authors use peptide TFL457?
4. The information about the amount of MTFL457 and the administration route in the in vivo study is missing.
5. Modified TFL457 (MTFL457) instead of TFL457 was used in the in vivo study. What is the difference between TFL457 and MTFL457? Will the modification alter the activity of the peptide in TrkB preservation?

1st Revision - authors' response

11 March 2019

Reviewer's comments (black) and author's responses (blue)

Referee #1 (Comments on Novelty/Model System for Author):

Unless the efficiency of transfection method used can be drastically improved the authors should rely on alternative methods such as the use of protease inhibitors affecting all cultured cells or else use model systems with all cultured cells expressing TrkB mutations thought to be targeted by proteases.

We apologize for the selection of the photograph presented in Figure 1C to illustrate peptide FITC-TMyc entry into neurons which might have driven the referee to misleading conclusions about the efficiency of peptide delivery. In these experiments, we observed most neurons in the culture stained with FITC-TMyc although there was some heterogeneity in the detected levels. So, in the selected picture we favored showing those neurons containing high peptide levels in detriment of those less stained. As stated by the referee, this is a central issue of our work. Thus, we have performed five independent experiments using biotin-labeled TMyc to precisely quantify peptide delivery (see details below). We believe that the new data presented strongly support our conclusions about the neuroprotective potential of this therapeutic approach.

Referee #1 (Remarks for Author):

The study by Tejada et al. examines an important and interesting issue related to TrkB signalling, and especially signalling duration. Exposing TrkB to its ligands including BDNF in particular has long been known to cause a rapid loss of cell surface, full-length TrkB. Much of the initial work in this area focussed on down-regulation as a result of ligand-mediated receptor internalisation. However, recent work by several groups begun to concentrate on additional mechanisms including proteolysis, especially in the context of

glutamate-mediated excitotoxicity. The new twist in the story is that increased calcium influx through the NMDA receptor may activate calpain(s) and preventing calpain activity has already been shown to significantly reduce TrkB proteolysis (see in particular Danelon et al MCN 2016, not quoted). Also, a recent study by Jeronimo-Santos actually mapped a calpain cleavage site on TrkB and indicated that various protease inhibitors reduce Abeta induced proteolysis of TrkB. For reasons that are not entirely clear given this context, the authors of the present study explored additional potential calpain cleavage sites on TrkB based on computer-based predictions. They then went on to design several peptides based on these predictions and coupled these to a cell permeable TAT sequence. These reagents were then used to explore a possible reduction of cytotoxicity exerted by NMDA on cultures neurons. The most promising peptide is also used in a test of local cortex infarction caused by photothrombosis.

While the topic is of considerable interest and so is the translational potential there are significant issues with the study.

We appreciate the reviewer's comments and have modified the Introduction (page 4) to emphasize the contributions that have established the importance of calpain activation for TrkB-FL regulation in excitotoxicity-associated conditions. In Vidaurre et al. (2012), we described TrkB dysregulation in human stroke and rat ischemia. We also demonstrated partial prevention of TrkB-FL processing by calpain inhibition in a cellular model of excitotoxicity, results soon confirmed by Gomes et al. (2012). The significance of calpain for TrkB-FL dysregulation has been also demonstrated in models of epilepsy (Danelon et al, 2016) or Alzheimer disease (Jeronimo-Santos et al., 2015). In that paragraph, we have also included another reference to the mapping by Jeronimo-Santos et al., (2015) of the calpain-processing site in TrkB-FL, in addition to those existing along the text. Altogether, we believe that the description of the current state of the art is now more accurate.

The first section in the Results (page 5) has also been revised to better explain the reasons behind the selection of the TrkB-FL sequences later included in the designed CPPs.

1. The most worrisome point is the use of cell-permeable peptides. As Fig. 1C illustrates only a fraction of the neurons is actually transfected. This fraction appears to be quite small. Also related to this important validation Figure, it is not clear what the arrows in the DAPI panel actually points to as nuclei are not clearly visible. Given the additional problem that the proportion of transfected cells may vary from experiment to experiment, the transfection strategy at the core of the study does not appear to be suitable for biochemical analyses such as those performed in the study. Obviously, the key assumption underlying mass analysis of cultured cells is that all or nearly all neurons are transfected and this condition is not fulfilled here. An additional problem is that according to Fig. 1C out of only three transfected cells one seem to be a non-neuronal cell.

Considering the importance of the point raised, we present new data analyzing peptide delivery into neurons using biotin-labeled TMyC (Bio-TMyC) (page 6, Figures 1C and EV2) which, in our experience, is more efficiently detected than FITC-TMyC. We have made five independent experiments and present in Figure 1C a representative high magnification image to show details of the peptide distribution into cell bodies and neurites of the majority of neurons. Additionally, Figure EV2 presents representative low magnification images that provides a more general idea about the efficiency of peptide delivery into neurons and supports the quantitation results. Using strict criteria to consider a neuron positive for Bio-TMyC staining, we analyzed a minimum of 100 neurons in each independent experiment. We found that the mean \pm SEM of neurons having internalized Bio-TMyC was $83 \pm 4\%$. We did not observe a high variability in the percentage of neurons containing peptide among the different experiments, a result in agreement with the consistency of the effects observed for TFL₄₅₇ in the several parameters measured along the various experiments presented.

2. Given the emphasis on calpain throughout it is difficult to understand why the calpain inhibitor used (Fig. 6) does not seem to be particularly effective, and apparently less so than a peptide designed on the basis of the assumption that calpain cleavage is at the core of TrkB proteolysis caused by exposure to NMDA. This also contrasts with the results of previous studies.

The point raised by the referee is important and needs to be further clarified. We do not find contradiction between previous studies and the modest effect observed for calpain inhibitors on excitotoxic neuronal death (Figure 6C). Several works have also shown that neuronal viability in excitotoxic conditions is only partially recovered by calpain inhibition (Rami et al., 1997; Gerencser et al., 2009; Wei et al., 2012). Calpain is central to cell physiology and, thus, inhibition of this enzyme might have dual effects in cells. In

fact, it has been recently shown that the major calpain isoforms in the brain have opposite roles in neuroprotection and neurodegeneration (Baudry and Bi, 2016). As reflected in the manuscript discussion, this has been an important reason encouraging the development of strategies to inhibit the action of calpain on precise sites of specific substrates involved in survival pathways, which are processed in the pathological conditions, instead of using generic calpain inhibitors (see for example Wu and Tymianski, 2018). References to these studies have been included in page 11 of the Results section and added to the bibliography.

3. The authors need to more explicitly acknowledge including in the Introduction that using a direct biochemical approach, a previous study clearly mapped including amino-terminal sequencing of a calpain fragment generated by cleavage of recombinant TrkB. Whilst this previous work is mentioned in Fig 1 (red arrow) and in the Discussion it is surprising that the authors chose to first describe their guess work about potential cleavage sites based on prediction programmes before acknowledging highly relevant, convincing previous work.

We have included an additional reference to recognize the important contribution made by Jeronimo-Santos et al., (2015) to determine the central role played by calpain in TrkB dysregulation in AD and experimentally establish the calpain-processing site in TrkB-FL juxtamembrane sequence.

4. Given the uncertainties related to the *in vitro* work aiming at validating the use of a particular peptide, the *in vivo* data are difficult to interpret, not least those related to dynamin-mediated endocytosis.

We hope that the new experiments presented about the efficiency of peptide entry into cultured neurons (Figure 1C and EV2) solve the referee's concerns about the relevance of the *in vitro* data. Altogether, the statistically significant effects found for TFL₄₅₇ in TrkB-FL stability, preservation of mRNAs and proteins key to neuronal survival and, moreover, prevention of both acute and chronic excitotoxicity, validate the neuroprotective potential of this peptide. From the *in vitro* model of excitotoxicity, we have moved forward to demonstrate the *in vivo* neuroprotective effects of TFL₄₅₇ by choosing a relevant stroke model. Our results prove that this peptide is delivered to the brain cortex and, in the ischemic brain, counteracts TrkB-FL downregulation, decreases infarct size and improves neurological outcome.

Considering the potential of TFL₄₅₇ for stroke therapy, we decided to investigate the primary mechanism of TFL₄₅₇ action. We could establish that this peptide prevented the decrease in levels of TrkB-FL present in the plasma membrane which was induced in cultured neurons subjected to excitotoxicity. One possibility was that TrkB-FL endocytosis might be a primary step for receptor processing induced by *in vitro* or *in vivo* excitotoxicity. Preliminary experiments in primary cultures treated with dynasore to inhibit dynamin-dependent endocytosis seemed to support such hypothesis. However, these results were not consistent, probably due to dynasore binding to serum proteins present in the culture medium (Preta et al. Cell Commun Signal. 2015 13:24). In contrast, we obtained statistically significant results *in vivo* after dynasore intraperitoneal injection proving that inhibition of endocytosis results in TrkB-FL stabilization after mice brain damage. These experiments are relevant and support that a primary mechanism of TFL₄₅₇ action in ischemia might be the interference of TrkB-FL endocytosis.

Referee #2 (Remarks for Author):

Ischemic stroke produces an infarct core surrounding an area called penumbra where neurons may subsequently die of excitotoxicity induced by overstimulation of the NMDA receptor. In the penumbra, BDNF expression is increased, but TrkB levels are reduced due to excitotoxicity-induced degradation. The authors hypothesize that preservation of TrkB in the penumbra would reduce the damage of stroke to the brain. They designed several cell-penetrating peptides (CPPs) containing TrkB intracellular sequences surrounding potential calpain cleavage sites. One CPP containing the TrkB juxtamembrane sequence (TFL457) was found to reduce TrkB degradation, preserve TrkB-PLC γ 1 signaling and increase neuronal viability in neuronal cultures after excitotoxicity. The authors further showed that the preservation effect of the peptide was due to increased levels of surface TrkB. Therefore, peptide TFL457 reduces TrkB endocytosis or enhances TrkB recycling back to the plasma membrane by sequestering proteins that normally interact with the TrkB juxtamembrane domain. Importantly, the authors showed that administration of peptide TFL457 reduced infarct size in a mouse ischemic model. These results are interesting, given that very few neuroprotective strategies are available to treat stroke. A few technical

issues should be addressed before the paper is published.

1. According to the authors' previous work, degradation of full-length TrkB should increase the abundance of tTrkB. This expected result is not consistently shown in Figure 1A.

In the revised manuscript, we present alternative experiments in Figures 2A and 2D which better represent the effects of NMDA treatment on TrkB isoforms as previously described (Vidaurre et al., 2012; Gomes et al., 2012). The discrepancy noted by the reviewer is probably due to difficulties intrinsic to the work with primary cultures of cortical neurons.

2. In Figure 5, the authors employed luciferase constructs containing a minimal promoter to show peptide TFL457 increases transcription of GluN1, GluN2A, BDNF and TrkB. Because the constructs do not have all regulatory elements, luciferase expression may not reflect the expression of endogenous genes. The results should be confirmed using real-time RT-PCR to measure levels of mRNAs from these genes.

Following the referee's recommendations, we have analyzed by RT-PCR the levels of GluN1, GluN2A, TrkB-FL and BDNF mRNAs in neurons subjected to excitotoxicity and compared them to those of the untreated neurons (Figure 5E). The results obtained are very relevant and strongly supportive of our conclusions. Thus, TFL₄₅₇ significantly reverts most changes in mRNA levels induced in the excitotoxic conditions as a consequence of the preservation of CREB and MEF2 regulatory activities. The results obtained for GluN2A suggest that, for this gene, TFs different from CREB and MEF2D might be also important for transcriptional regulation in excitotoxicity. However, levels of protein GluN2A are still partially recovered in excitotoxic neurons pretreated with TFL₄₅₇, suggesting that additional mechanisms might enhance translation of GluN2A mRNA or increase the stability of the encoded protein.

3. The authors used NA-1 peptide to show that the peptide is able to get into the brain. Why did not the authors use peptide TFL457?

These experiments were performed in the initial stages of this research project, in order to confirm that Tat peptides could be delivered to the cortex. Therefore, we used a well-described model CPP, NA-1, which had been established to be neuroprotective in different stroke models and extensively described in the literature (Aarts et al., 2002; Li et al., 2007, PMID: 17360906; Sun et al., 2008, PMID: 18617669; Bratane et al., 2011, PMID: 21903963; Cook et al., 2012, PMID: 22388811; Teves et al., 2015, PMID: 26661213).

Once designed TFL₄₅₇, we presumed that, having the same cell-penetrating Tat sequence than NA-1, that peptide would be also able to cross the BBB. This conclusion was supported by the results obtained in the mice model of ischemia showing that TFL₄₅₇ counteracts TrkB-FL downregulation and reduces infarct volume and neurological damage. However, delivery of the neuroprotective peptide into the brain is a critical point of the proposed therapy. Thus, we have now specifically analyzed entry of biotin-labeled TFL₄₅₇ into cortical neurons and present these data in Figure 7A and page 13 to better support our *in vivo* findings. The introductory results obtained for NA-1 are now presented in Figure EV4.

4. The information about the amount of MTFL457 and the administration route in the *in vivo* study is missing.

This information is in page 19 of the revised manuscript (former page 20) in the section Material and Methods. CPPs (10 nMole/g; >95% purity; GenScript) were retro-orbitally injected in the venous sinus in a single dose 10 min after damage initiation.

5. Modified TFL457 (MTFL457) instead of TFL457 was used in the *in vivo* study. What is the difference between TFL457 and MTFL457? Will the modification alter the activity of the peptide in TrkB preservation?

The only difference between MTMyc or MTFL₄₅₇ and their corresponding unmodified versions is that, in the former, the N-ter and C-ter aminoacids are respectively acetylated and amidated. This information is in page 19 of the revised manuscript (former page 20) and Table 1, and now has been also included in the Results section (page 13). Compared to the free α -amino and carboxyl groups present in the N-ter and C-ter of unmodified peptides, it has been described that these modifications mimicking the natural protein structure improve the plasma stability of injected peptides.

As demonstrated in Figure 7C and D, MTF_{L457} significantly stabilized TrkB-FL levels in the ischemia model. We now present a new supplementary figure to show that MTF_{L457} similarly interferes TrkB-FL processing in the *in vitro* model of excitotoxicity (Figure S4).

2nd Editorial Decision

4 April 2019

Thank you for your email of yesterday.

I would like to invite you to revise your article along the lines that you indicated in your letter. This revision will therefore consists of minor text changes and addition of available data as you indicated. Please make sure to upload a point-by-point letter to answer the referee's concerns.

Please submit your revised manuscript as soon as possible.

I look forward to reading a new revised version of your manuscript as soon as possible.

***** Reviewer's comments *****

Referee #1 (Comments on Novelty/Model System for Author):

Whilst the new experiments clarify key issues related to the proportion of cultured neurons targeted by the peptides used there is still room for improvement and a critical methodology point needs attention (see below).

Referee #1 (Remarks for Author):

As previously pointed out this is a valuable contribution. Yet it is still in need of revision.

1. As there are no indications that the neurons were grown in the presence of BDNF one wonders what the origin of the P-Tyr signal is in Fig 3? These results are not easy to reconcile with the data presented in Annex S2 that clearly show a BDNF-dependent phosphorylation of TrkB. If BDNF has not been added throughout as the Methods suggest (see above) it would be important to also illustrate the results of excitotoxicity prevention by exogenous BDNF given that this notion is central to the work presented here (see Introduction). As the authors rightly argue, there is an inherent variability to primary cultures and the addition of the presumed rescue effects of BDNF following NMDA and glycine addition may help calibrate the results in the context of the results obtained by others.

2. The text is not always sufficiently clear. Example are "This CPP" at the beginning of the Discussion and the statement that "Little is known about how NMDAR overactivation affects TrkB", a statement that is directly contradicted by the revised Introduction indicating that quite a lot is already known following work by the authors as well as by other groups. Also at the beginning of the Discussion the use of "prove" should be avoided.

3. Previous work by Gamir-Moralla and colleagues (including some of the authors of the present submission) should be more explicitly mentioned and discussed. Indeed, the strategy used was very similar and neurons survival also assessed following NMDA addition treated with a cell-permeable peptide targeting Kidins220/ARMS, also investigated as a calpain target. The authors should discuss the relative merits of the two approaches and compare the results in terms of excitotoxicity prevention.

2nd Revision - authors' response

9 April 2019

Referee #1 (Comments on Novelty/Model System for Author):

Whilst the new experiments clarify key issues related to the proportion of cultured neurons targeted by the peptides used there is still room for improvement and a critical methodology point needs attention (see below).

Referee #1 (Remarks for Author):

As previously pointed out this is a valuable contribution. Yet it is still in need of revision.

Comment #1

As there are no indications that the neurons were grown in the presence of BDNF one wonders what the origin of the P-Tyr signal is in Fig 3?

Cultures of dissociated cortical neurons present spontaneous neuronal/synaptic activities (see Pasquale et al., 2017 and references therein) which could potentiate basal TrkB tyrosine kinase activity by a mechanism that requires calcium-influx through NMDARs and calcium channels (Du et al., 2003). Therefore, spontaneous neuronal activity might explain the low levels of TrkB phosphorylation observed in non-stimulated cultures (Appendix Figure S1A, lower panels, and Figure 3), results similar to those obtained before by other groups (see, for example, Fig. 1G, Du et al., 2003 or Fig. 4A, Gomes et al., 2012) and also us (Fig. 3b, Vidaurre et al., 2012). A short phrase has been added to explain these results (page 8), and new references have been included in the bibliography.

These results are not easy to reconcile with the data presented in Annex S2 that clearly show a BDNF-dependent phosphorylation of TrkB.

Different groups have demonstrated BDNF-dependent TrkB phosphorylation upon neurotrophin treatment in cellular models where basal receptor phosphorylation was observed in the non-stimulated cultures (Fig. 1G, Du et al., 2003; Fig. 4A, Gomes et al., 2012; Fig. 3b, Vidaurre et al., 2012). The images presented in Appendix Fig. S1 tried to highlight the activation of TrkB phosphorylation by BDNF. Longer exposures of the same panels have been now included to also show basal receptor phosphorylation in non-treated cultures, in agreement with results in Figure 3B.

If BDNF has not been added throughout as the Methods suggest (see above) it would be important to also illustrate the results of excitotoxicity prevention by exogenous BDNF given that this notion is central to the work presented here (see Introduction). As the authors rightly argue, there is an inherent variability to primary cultures and the addition of the presumed rescue effects of BDNF following NMDA and glycine addition may help calibrate the results in the context of the results obtained by others.

The *in vitro* model of excitotoxicity developed by Dr. Choi consists of mixed cultures of neuronal and glial cells, main populations in the brain cortex. Primary cultures, grown in MEM supplemented with 5% horse serum and 5% FBS for 12-14 DIVs for neuronal maturation, are robust since astrocytes (approx. 20% of total cells) reduce glutamate levels in the extracellular media. Thus, upon addition of high concentrations of glutamate or NMDA, strong excitotoxic responses are induced in the mature neurons while the viability of the astrocyte subpopulation is not affected (Choi 1985; Choi et al., 1987).

It has been established that, in this model, neuronal death induced by excitotoxicity occurs by necrosis, both if rapidly (Choi 1987 PMID: 2880938) or slowly triggered (Gwag et al., 1997), and cannot be prevented by BDNF treatment. In contrast, BDNF is able to attenuate neuronal apoptotic death induced by serum deprivation or other stimulus. Our experiments confirm that BDNF cannot prevent neuronal death induced by excitotoxicity in our primary cultures or interfere TrkB-FL processing induced by NMDA. These results are now presented in panels B and C of Appendix Fig. S1.

These *in vitro* results might reflect those obtained in the mice model of ischemia (Fig. 7), where strong cortical neurodegeneration and decreased TrkB-FL levels take place in the infarct in the presence of endogenous BDNF. They are also in accordance with studies mentioned in the manuscript's introduction rejecting the involvement of BDNF in stroke recovery (Hirata et al., 2011).

Alternative models of excitotoxicity are based on semi-pure hippocampal neurons grown in NB medium with B27 (containing approximately 3% of glial cells) where treatment with NMDA activates apoptotic mechanisms (Almeida et al., 2005). In this system, BDNF could not protect neurons subjected to brief excitotoxic stimulation if added after the insult but had a protective effect when incubated together with (Fig. 8A, Gomes et al., 2012) or before NMDA addition (Lau et al., 2015).

It has been shown before that the type of neuronal death induced in cell culture by glutamate might be necrotic or apoptotic (Ankarcrone et al., 1995), a situation similarly observed *in vivo* in the ischemic tissue.

We have included this topic in the discussion.

Comment #2

The text is not always sufficiently clear. Example are "This CPP" at the beginning of the Discussion and the statement that "Little is known about how NMDAR overactivation affects TrkB", a statement that is directly contradicted by the revised Introduction indicating that quite a lot is already known following work by the authors as well as by other groups. Also at the beginning of the Discussion the use of "prove" should be avoided.

We appreciate the advice about the text and more concise phrases have been now included in the highlighted sections.

Comment #3

Previous work by Gamir-Moralla and colleagues (including some of the authors of the present submission) should be more explicitly mentioned and discussed. Indeed, the strategy used was very similar and neurons survival also assessed following NMDA addition treated with a cell-permeable peptide targeting Kidins220/ARMS, also investigated as a calpain target. The authors should discuss the relative merits of the two approaches and compare the results in terms of excitotoxicity prevention.

The discussion has been modified to include more information about our previous work with Tat-K, performed in the cellular model of excitotoxicity, and the potential of this peptide if demonstrated efficient also *in vivo*.

3rd Editorial Decision

29 April 2019

Thank you for the submission of your revised manuscript to EMBO Molecular Medicine. We have received feedback from the referee #1 who is now satisfied with the revision. I am pleased to inform you that we will be able to accept your manuscript pending final editorial amendments.

3rd Revision - authors' response

2 May 2019

Authors made the requested changes.

Corresponding Author Name: Margarita Díaz-Guerra

Manuscript Number: EMM-2018-09950